# Uncoupling of invasive bacterial mucosal immunogenicity from pathogenicity

Simona P. Pfister [1,2,10], Olivier P. Schären [1,2,10], Luca Beldi [1], Andrea Printz[1], Matheus D. Notter[1,2], Mohana Mukherjee [1,2], Hai Li [3], Julien P. Limenitakis[3], Joel P. Werren [1,2], Disha Tandon[1,2], Miguelangel Cuenca[1], Stefanie Hagemann[1], Stephanie S. Uster[1], Miguel A. Terrazos [1], Mercedes Gomez de Agüero [3], Christian M. Schürch [4,5,6], Fernanda M. Coelho [1], Roy Curtiss III[7], Emma Slack [8], Maria L. Balmer [9] & Siegfried Hapfelmeier [1✉]

There is the notion that infection with a virulent intestinal pathogen induces generally stronger mucosal adaptive immunity than the exposure to an avirulent strain. Whether the associated mucosal inflammation is important or redundant for effective induction of immunity is, however, still unclear. Here we use a model of auxotrophic *Salmonella* infection in germ-free mice to show that live bacterial virulence factor-driven immunogenicity can be uncoupled from inflammatory pathogenicity. Although live auxotrophic *Salmonella* no longer causes inflammation, its mucosal virulence factors remain the main drivers of protective mucosal immunity; virulence factor-deficient, like killed, bacteria show reduced efficacy. Assessing the involvement of innate pathogen sensing mechanisms, we show MYD88/TRIF, Caspase-1/Caspase-11 inflammasome, and NOD1/NOD2 nodosome signaling to be individually redundant. In colonized animals we show that microbiota metabolite cross-feeding may recover intestinal luminal colonization but not pathogenicity. Consequent immunoglobulin A immunity and microbial niche competition synergistically protect against *Salmonella* wild-type infection.

---

[1] Institute for Infectious Diseases, University of Bern, Bern, Switzerland. [2] Graduate School GCB, University of Bern, Bern, Switzerland. [3] Maurice Müller Laboratories (DBMR), Universitätsklinik für Viszerale Chirurgie und Medizin (UVCM) Inselspital, Bern, Switzerland. [4] Institute of Pathology, University of Bern, Bern, Switzerland. [5] Institute of Pathology and Neuropathology and Comprehensive Cancer Center, University Hospital Tübingen, Tübingen, Germany. [6] Baxter Laboratory for Stem Cell Biology, Department of Microbiology and Immunology, Stanford University School of Medicine, Stanford, CA, USA. [7] Biodesign Institute, Arizona State University, Tempe, AZ, USA. [8] Institute for Food, Nutrition and Health, D-HEST, ETH Zürich, Switzerland. [9] Department of Biomedicine, Immunobiology, University of Basel, Basel, Switzerland. [10] These authors contributed equally: Simona P. Pfister, Olivier P. Schären. ✉email: siegfried.hapfelmeier@ifik.unibe.ch

Mounting a functional anti-microbial adaptive immune response depends on concomitant induction of an innate immunogenic response through pattern recognition receptor (PRR) activation[1]. PRRs sense conserved microbial molecular structures, such as bacterial lipopolysaccharide (LPS), peptidoglycan, and flagellin, that are conserved across pathogenic and non-pathogenic microorganisms[2]. Pathogen-specific virulence factors such as type 3 secretion system (T3SS) components[3] and intracellular toxin action have also been shown to be specifically sensed by PRRs[4,5]. The integration of diverse PRR signals is believed to regulate immune responses according to the nature of the microbial threat[6,7]. Natural and artificial PRR signaling agonists are consequently exploited pharmaceutically as pro-immunogenic additives or adjuvant components of vaccines[8]. Besides the immunogenic response, PRR activation by pathogens may also drive inflammation and innate anti-microbial defense. This arm of the innate immune system is important for the control of primary pathogen infection, but is also responsible for the adverse effects of inflammation and defense that damage host tissue and symbiotic microbiota, which may be exploited by some mucosal pathogens[9].

The intestinal mucosal membranes are colonized continuously with a diverse symbiotic microbiota and are guarded by a complex mucosal immune system. The mucosa is well adapted to stable symbiosis with non-pathogenic microbes. Multiple physical and chemical barriers as well as active immune tolerance avoid the unnecessary activation of immune defense mechanisms by harmless symbiotic microbes or food antigens[10]. Only virulent mucosal pathogens normally induce inflammatory responses. Avirulent, fully attenuated pathogens are inefficient at driving inflammation, but also tend to induce less effective adaptive immunity than virulent pathogens[11,12]. It is consequently difficult to induce protective mucosal immunity safely with adequately attenuated live vaccines—this compromises vaccination efforts in developing countries for which safe, effective, and easy-to-administer oral vaccines are urgently needed[13,14].

While there is a clear difference between the immune responses induced by virulent and non-virulent variants of a pathogenic bacterium, it is unclear which aspects of bacterial virulence may be differentially sensed by the immune system to induce efficacious adaptive immunity. Virulence factors enable pathogens to colonize privileged body sites, overgrow host defenses, and consequently damage host tissue architecture and function. Inactivated (killed) pathogenic microbes are avirulent, because they are sterile and most virulence mechanisms (apart from, for example, stable exotoxins) are dependent on bacterial viability. Our question was whether a pathogen that combines sterility and viability, which expresses molecularly functional virulence factors in vivo but is unable to replicate, still retains its mucosal immunogenicity.

To address this question, we apply a quantitative Salmonella enterica serovar Typhimurium (STm) infection model in germ-free mice in which live bacterial replication in vivo is blocked. We use auxotrophic mutants of STm (STm$^{Aux}$) that are genetically engineered to be fully replication incompetent in germ-free animals and host tissues. As we previously established in non-pathogenic enterobacteria[15,16], STm$^{Aux}$ colonization in germ-free mice is limited by the quantity of the bacterial inoculum and fully transient, allowing the germ-free host to return to germ-free status.

Using germ-free mice, this experimental approach allows us to rigorously test the following issues. First, whether the mucosal immunogenic response can distinguish between virulence factor proficient and deficient intestinal bacteria also in the absence of an acute inflammatory response and pathology. Secondly, whether the remaining immunogenic response would depend on

similar PRR signaling pathways as the innate immune defense. These fundamental studies are carried out in a germ-free setting, to avoid the possible confounding effect of auxotrophic metabolite crossfeeding by bacteria of the gut microbiota in vivo. Extending our results into colonized mice, we move on to show that indeed crossfeeding by the microbiota can recover efficient intestinal colonization, but not pathogenicity of STm$^{Aux}$. Strictly confined by the mucosal barrier it then combines virulence factor-dependent immunogenicity and avirulence with the added benefit of pathogen niche competition.

## Results

**Proliferation-incompetent STm$^{Aux}$ induces functional immunity.** Mucosal tissue invasion and virulence of STm are mediated by two type 3 secretions systems (T3SS) encoded on Salmonella pathogenicity islands, SPI1 and SPI2 (refs. [3,17–19]). Activity of the SPI1-encoded T3SS induces early mucosal inflammation[20,21]. As the invading and tissue-overgrowing virulent bacteria responsible are subject to pronounced population bottle necks[3,22], we hypothesized that a strain of live STm encoding functional virulence factors would retain its invasiveness with associated adaptive immunogenicity, despite being unable to replicate and overall avirulent.

To test this hypothesis we generated an auxotrophic STm strain (STm$^{Aux}$) that strictly requires supplementation with the essential peptidoglycan constituents D-alanine (D-Ala) and meso-diaminopimelic acid (m-Dap) to grow and survive cell division (Supplementary Fig. 1A, B). Like the homologous model in commensal Escherichia coli developed previously[15,16], STm$^{Aux}$ colonized the gastrointestinal tract of germ-free mice only transiently, allowing rapid and full recovery to germ-free status, as neither host metabolism nor diet could substitute the auxotrophic requirement for these metabolites (Fig. 1a, b). Salmonella T3SS competence or deficiency had no effect on STm$^{Aux}$ colonization kinetics. Bacterial quantitation in small intestinal (Supplementary Fig. 2A) and cecal (Supplementary Fig. 2B) contents at early time points revealed small intestinal transit of STm$^{Aux}$ in quantities similar to wild-type STm until 2.5 h following inoculation. At 4.5 h, STm$^{Aux}$ had transited from the small intestine into cecum without evidence for replication (Supplementary Fig. 2A, B; compare STm$^{Aux}$ numbers between small intestine at 2.5 h and cecum at 4.5 h), whereas wild-type STm populations had begun to expand in the cecum. By 34 h after inoculation wild-type STm stably colonized all intestinal segments, whereas STm$^{Aux}$ densities had sharply declined. No spontaneous D-Ala/m-Dap-independent revertants have been isolated ex vivo during these experiments.

D-Ala/m-Dap auxotrophic bacteria depleted of D-Ala or m-Dap, analogous to wild-type bacteria exposed to beta-lactam antibiotics, remain active until self-destruction by programmed autolytic cell death occurring at the onset of cell division[16,23]. Accordingly, D-Ala- and m-Dap-depleted STm$^{Aux}$ displayed normal cell invasiveness, as demonstrated by immunofluorescence microscopy and gentamicin protection assay (Fig. 1g, h). In germ-free mice, following enteral administration of $10^{10}$ colony-forming units (CFU) of STm$^{Aux}$ by gavage, the invasive auxotroph was found to be completely avirulent. In contrast to wild-type STm, STm$^{Aux}$ was rarely recoverable from mesenteric lymph nodes (mLN), liver, or spleen (Fig. 1f). It no longer induced detectable levels of typhocolitis (inflammation of the cecum, the main enteric histopathology in the non-typhoidal invasive salmonellosis mouse model[24]) as determined either by quantification of cecal luminal inflammation marker lipocalin-2 (Fig. 1c) or by histopathologic scoring (Fig. 1d, e). Quantification of early mRNA markers of chemokine and other innate activation

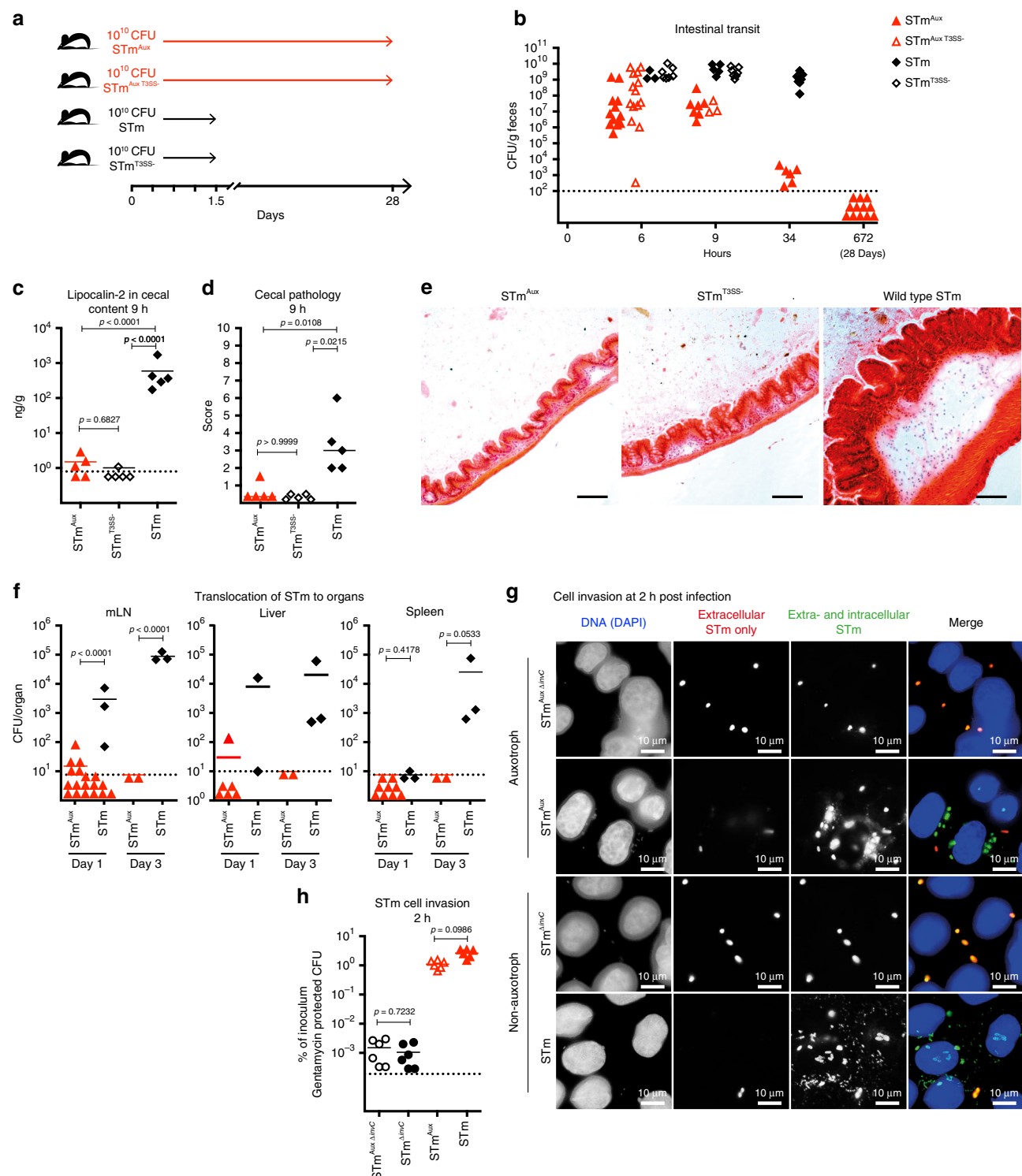

signals in total cecum tissue supported the conclusion that STm[Aux] is avirulent (Supplementary Fig. 2C).

Transitory intestinal mucosal conditioning by live STm[Aux] bacteria (Fig. 2a) induced an adaptive immune response highly protective against the re-challenge of the germ-free animals with non-auxotrophic wild-type STm. While immunity had no effect on the large intestinal luminal load of the challenge strain (Fig. 2b), it protected against its intestinal pathogenesis (Fig. 2c, d) and limited penetration to the mLNs, liver, and spleen (Fig. 2e). Protective immunity was associated with high STm-

specific titers of intestinal secretory IgA measured by live bacterial flow cytometry[25] (Fig. 2f, g), and was abolished in B cell- and antibody-deficient $J_H^{-/-}$ mice (Fig. 2h–j). B and T cell-deficient RAG-deficient mice phenocopied $J_H$-deficient mice (Supplementary Fig. 3A–D). Hence, B cell immunity is functionally required for STm[Aux]-induced intestinal protective immunity. The live STm[Aux] dose–response relationship was examined by comparing the mucosal conditioning with doses of $10^{10}$, $10^8$, and $10^6$ live STm[Aux], which revealed that induction of functional immunity required doses greater than $10^8$ live STm[Aux] (see extended dataset

**Fig. 1 Transient colonization of GF mice with STm$^{Aux}$. a** Mice were inoculated at day 0 with $10^{10}$ CFU of either auxotrophic ($^{Aux}$; red symbols) or non-auxotrophic control (black symbols) STm strains that were either type 3 secretion competent (STm/STm$^{Aux}$, filled symbols) or isogenic type 3 secretion-deficient mutants (STm$^{T3SS-}$/STm$^{Aux\ T3SS-}$, open symbols). **b** Time course of viable bacteria of each strain recoverable from feces (STm$^{Aux}$ $n = 32$, STm$^{Aux\ T3SS-}$ $n = 15$, STm$^{T3SS-}$ $n = 12$, STm $n = 11$, animals examined over nine independent experiments). **c** Lipocalin-2 concentration in cecal contents at 9 h after inoculation (STm$^{Aux}$ $n = 5$, STm$^{T3SS-}$ $n = 5$, STm $n = 5$ animals). **d** Cecal histopathology score at 9 h after inoculation. Each symbol represents one individual (STm$^{Aux}$ $n = 5$, STm$^{T3SS-}$ $n = 5$, STm $n = 5$ animals). **e** Cecal histology at 9 h after inoculation with indicated STm strains. H&E staining. Scale bar: 100 μm (STm$^{Aux}$ $n = 5$, STm$^{T3SS-}$ $n = 5$, STm $n = 5$ animals). **f** Organ loads of T3SS-proficient STm$^{Aux}$ and STm in mLN, liver, and spleen on day 1 (mLN: STm$^{Aux}$ $n = 18$, STm $n = 3$; liver: STm$^{Aux}$ $n = 6$, STm $n = 2$; spleen: STm$^{Aux}$ $n = 9$, STm $n = 3$ animals examined over four independent experiments) and 3 (mLN: STm$^{Aux}$ $n = 2$, STm $n = 3$; liver: STm$^{Aux}$ $n = 2$, STm $n = 3$; spleen: STm$^{Aux}$ $n = 2$, STm $n = 3$ animals examined over two independent experiments) post inoculation. **g** Immunofluorescence of HeLa cells infected for 2 h with wild type (STm), SPI1 T3SS-deficient (STm$^{\Delta invC}$), auxotrophic SPI1 T3SS-proficient (STm$^{Aux}$), and auxotrophic SPI T3SS-deficient (STm$^{Aux\ \Delta invC}$) STm. Cells were stained with DAPI (DNA/nuclei, blue), and with anti-STm group B antiserum and labeled secondary antibodies consecutively before, and after membrane permeabilization to differentiate extracellular (red + green) and intracellular (green only) STm. Scale bar: 10 μm (six samples were examined over two independent experiments for each condition). **h** Quantification of gentamicin-protected intracellular STm$^{Aux\ \Delta invC}$ (black open circles, $n = 6$ wells examined over two independent experiments), STm$^{\Delta invC}$ (black filled circles, $n = 6$ wells examined over two independent experiments), STm$^{Aux}$ (red open triangles, $n = 6$ wells examined over two independent experiments), and STm$^{\Delta invC}$ (red filled triangles, $n = 6$ wells examined over two independent experiments) in HeLa cells 2 h after infection. Statistics: bars indicate mean (**c, f, h**) or median (**d**) values. Horizontal dotted lines indicate the lower limit of detection (**b, c, f, h**). Panel **c** was analyzed with ordinary one-way ANOVA and Tukey's test for multiple comparison. Panel **d** was analyzed with a two-sided multicomparison Kruskal–Wallis test and Dunn's post hoc test. Panel **f** was analyzed with unpaired two-tailed $t$-test for each day. Panel **h** was analyzed with two-way ANOVA (virulence and auxotrophy as the two factors) and Sidak multiple comparison correction. Source data are provided as a Source Data file. Detailed statistical metrics are available in the Supplementary Statistical Analysis file.

in Supplementary Fig. 4A–G). Thus, STm$^{Aux}$ allowed us to probe mucosal immunity in a strictly dose-dependent manner. This data showed the threshold effects of STm$^{Aux}$ conditioning in germ-free mice, which would not be achievable with conventional non-auxotrophic bacteria that would exponentially expand rapidly to reach high intestinal densities independently of inoculum size.

**Optimal protective efficacy of STm$^{Aux}$ is viability dependent.** We next addressed how relevant STm$^{Aux}$ viability is for the induction of functional intestinal immunity. The replication incompetency of live STm$^{Aux}$ in germ-free mice allowed us to quantitatively compare the functional effects of mucosal exposure to live versus killed STm: both live and killed STm$^{Aux}$ cells are sterile entities in germ-free mice. Parallel groups of germ-free mice were intestinally conditioned by gavage with STm$^{Aux}$ inocula administered either live or following inactivation by peracetic acid (PAA) treatment. PAA killing is highly effective and has been shown to preserve mucosally protective STm surface B cell epitopes[26,27]. Naïve germ-free animals served as negative controls. Four weeks after the first treatment, the germ-free animals of all three groups were challenged orally with virulent wild-type STm and studied at days 1 and 4 after the challenge, respectively (Fig. 3a). Compared to live STm$^{Aux}$-conditioned mice, PAA-killed STm$^{Aux}$-induced STm-specific IgA titers were reduced at day 1 of challenge (Fig. 3b, Supplementary Fig. 5). Yet, by day 4 this difference was no longer apparent. However, while pretreatment with either PAA killed or live STm$^{Aux}$ were similarly protective against early wild-type STm-induced mucosal inflammation at day 1 after challenge, only live STm$^{Aux}$ preconditioning provided effective protection from intestinal pathology and organ infection until day 4 (Fig. 3c–g). Notably, live STm$^{Aux}$-induced immunity not merely delayed the onset of disease, but protected the germ-free mice from lethal STm infection. Live STm$^{Aux}$-conditioned germ-free mice that were followed up for 3 weeks following challenge remained free of macroscopic evidence of severe infection and were recovering at the time of sacrifice (Supplementary Fig. 6). None of these effects were explained by differences in fecal or cecal luminal colonization levels of the challenge strain, which were similar across all experimental groups (Fig. 3h, i, Supplementary Fig. 6B).

**Salmonella type 3 secretion signifies robust immunogenicity.** We next asked whether or not the viability-dependency of functional mucosal immunogenicity of STm$^{Aux}$ is virulence factor related. We hypothesized that host interaction through *Salmonella* T3SSs (whose function is energy and viability dependent) signifies the functional immunogenicity of live STm$^{Aux}$. If this was true, T3SS deficiency would diminish the mucosal efficacy of live STm$^{Aux}$.

We tested this hypothesis by comparing the protective effect of the enteral conditioning of germ-free mice with matching doses of live T3SS-competent and isogenic T3SS-deficient mutant strains of STm$^{Aux}$ (STm$^{Aux\ T3SS-}$). Two different isogenic STm$^{Aux\ T3SS-}$ mutants were tested: a complete SPI1 and SPI2 genomic island deletion mutant (ΔSPI1 ΔSPI1) devoid of T3SS genes entirely[28], and a Δ*invC* Δ*ssaV* mutant expressing defective T3SSs[29,30]. Mice treated with equivalent doses of PAA-killed STm$^{Aux}$ or naïve mice served as controls. Four weeks after the first treatment, the germ-free animals were enterally challenged with wild-type STm, and studied at days 1 and 3 after challenge (Fig. 4a). Analysis of the severity of challenge infection and mucosal pathology at day 3 revealed that live, T3SS-deficient STm$^{Aux}$ strains induced less robust functional protective immunity than T3SS-competent STm$^{Aux}$, and their efficacy against intestinal mucosal pathology (Fig. 4b, c) and bacterial penetration to mLN (Fig. 4d) was no longer significantly better than that of PAA-killed STm$^{Aux}$. Genetic deletion of the three most important SPI1 effector protein genes (*sopE*, *sopE2*, and *sipA*) required for early SPI1 T3SS-mediated intestinal STm pathogenesis[31] also resulted in reduced efficacy (Supplementary Fig. 7). This suggests that not merely immune recognition of a functional T3SS apparatus but rather mucosal pathogenesis-related type 3 effector protein functions are driving the superior immunogenicity of the T3SS-competent STm$^{Aux}$ strain.

As in the previous experiment (Fig. 3b) killed STm$^{Aux}$- as well as STm$^{Aux\ T3SS-}$-preconditioned mice displayed reduced STm-specific IgA titers at day 1 of challenge (Fig. 4e, Supplementary Fig. 8). Yet, by day 3 mice of all three treatment groups had similar intestinal IgA titers. Immunoglobulin repertoire sequencing analysis of small intestine and mLN revealed overlapping IgA repertoires following mucosal conditioning with live T3SS-competent STm$^{Aux}$- versus T3SS-incompetent STm$^{Aux}$ that clustered separately from those of naïve germ-free control mice

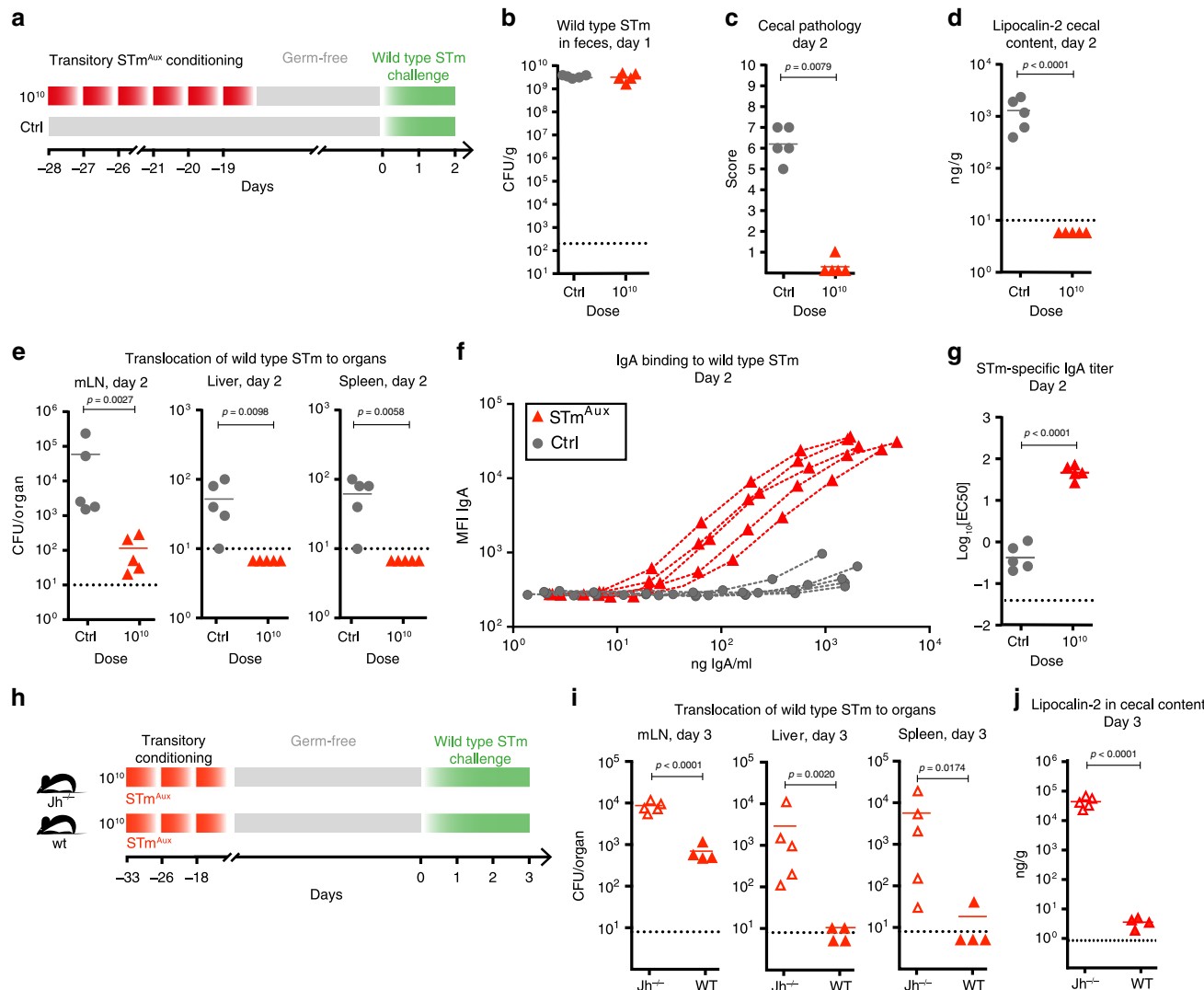

**Fig. 2 Intestinal conditioning of GF mice with STm^Aux induces B cell dependent functional intestinal immunity. a** Germ-free mice were enterally conditioned with six doses of $10^{10}$ CFU of STm^Aux (red triangles, n = 5 animals) or were left untreated (gray-filled circles, n = 5 animals). Four weeks after the first treatment (day 0) mice were challenged with wild-type STm ($10^5$ CFU) and analyzed 2 days later. Each symbol represents one individual. **b** Shedding of wild-type STm in feces 1 day after challenge. **c** Cecal histopathology score at day 2 after challenge. Each symbol represents one individual. **d** Lipocalin-2 concentration in cecal contents at day 2 after challenge. **e** Bacterial burden of wild-type STm in mLNs, spleens, and livers at day 2 after challenge. **f** Intestinal secretory IgA was isolated at day 2 after challenge. IgA binding to wild-type *Salmonella* was quantified at different antibody concentrations by live bacterial flow cytometry. Connected symbols represent one individual. **g** STm-specific titer ($-\log EC_{50}$) calculated from the STm-IgA titration curve plotted in Fig. S4J. **h** Experimental design: Germ-free Jh$^{-/-}$ (open symbols, n = 5 animals) and wild-type control mice (filled symbols, n = 4 animals) were enterally conditioned three times with $10^{10}$ CFU of live STm^Aux. Thirty-three days after the first treatment (day 0) all mice were challenged with of wild-type STm ($10^3$ CFU). **i** Bacterial burden of wild-type STm in mLNs, spleens, and livers at day 3 after challenge. **j** Lipocalin-2 concentration in cecal contents at day 3 after challenge. Statistics: bars indicate mean (**b**, **d**, **e**, **g**, **i**, **j**) or median (**c**) values. Horizontal dotted lines indicate the detection limit. Panel **c** was analyzed with a two-sided Mann–Whitney U-test. Panels **d**, **e**, **g**, **i** and **j** were analyzed with the unpaired two-tailed t-test. Source data are provided as a Source Data file. Detailed statistical metrics are available in the Supplementary Statistical Analysis file.

(Supplementary Fig. 9). Repertoire overlap was measured by calculating the geometric mean of relative overlap frequencies between CDR3 amino acid sequence usage (see Methods section). Preprocessed clonotype amino acid sequences and metadata description are available as supplementary data files (Supplementary Data 1–21).

O-serotype specific IgA has been shown previously to be a necessary component of any intestinal immune protection induced by killed or live STm[12,26,27]. O-antigen is a dominant polysaccharide antigen and in binding assays tends to mask other surface epitopes from antibody recognition, which is the basis of O-serotyping. To specifically study O-serotype-independent

*Salmonella* surface binding IgA, germ-free mice were preconditioned with STm^Aux but challenged with the different *Salmonella* serotype Enteritidis (SEn) (Supplementary Fig. 10A). The resulting intestinal IgA had reduced surface reactivity towards O-antigen-deficient (rough) STm compared to wild-type (smooth) STm, as expected (Supplementary Fig. 10B, C). However, the non-O-antigen-specific IgA cross-reacted between rough STm and rough SEn (Supplementary Fig. 10B, C). It also cross-reacted with smooth wild-type SEn, suggesting that it contributes to serotype-independent *Salmonella* surface reactivity (Supplementary Fig. 10B, C). Although the O-serotype-independent IgA component alone is insufficient[12,26,27], it may complement

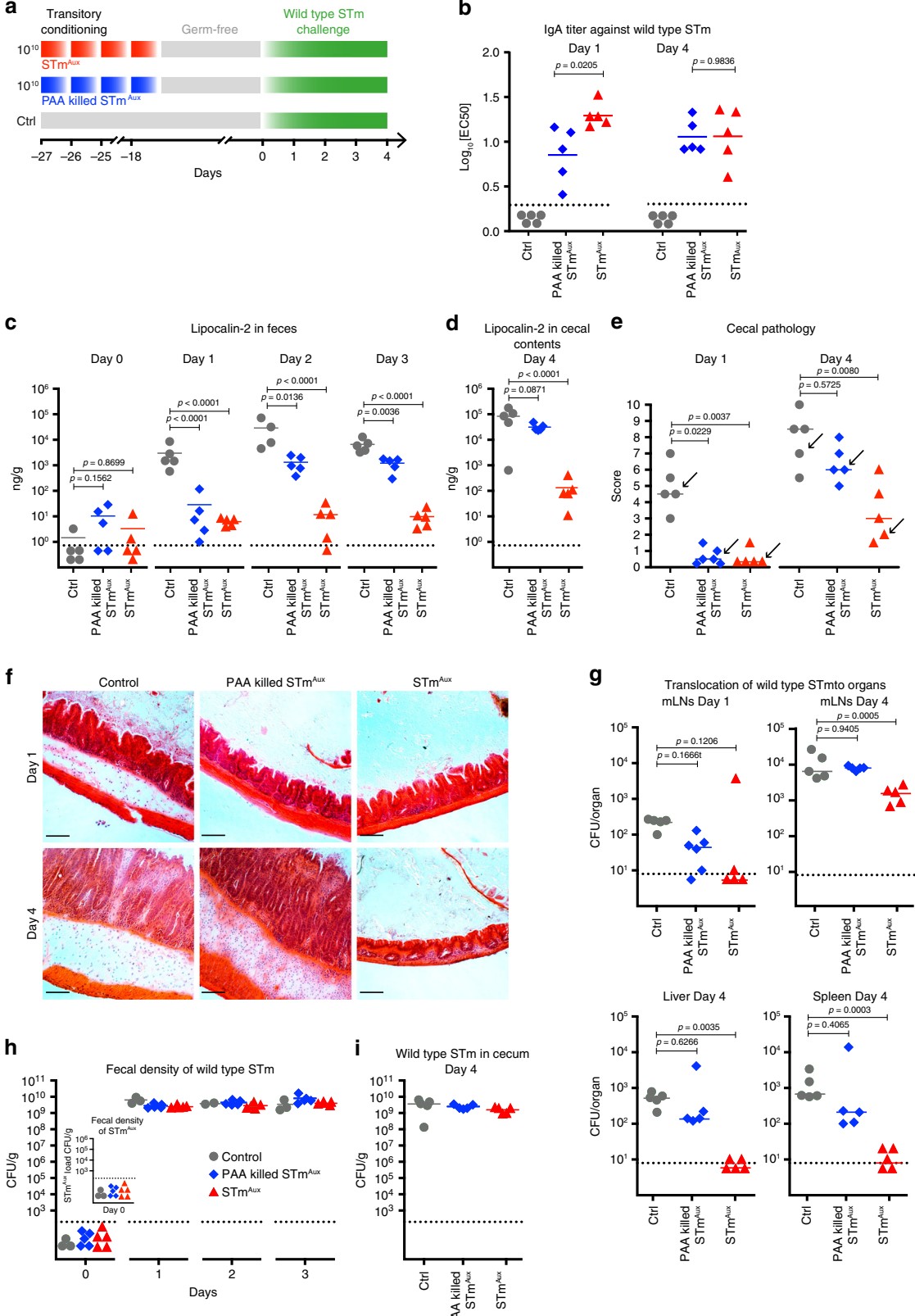

O-antigen-specific IgA in protective mucosal immunity. Supporting this idea, we found that, although both killed and T3SS-deficient STm^Aux preconditioning at day 3 of challenge resulted in robust IgA titers towards smooth STm (Fig. 4e, panel Day 3), IgA binding to rough STm was significantly reduced (Fig. 4f).

These data show that *Salmonella* T3SS-dependent virulence functions signify the mucosal immunogenic efficacy of life STm^Aux in absence of inflammation. The underlying T3SS-dependent IgA B cell response is characterized by a less O-antigen-restricted bacterial surface reactivity.

**Fig. 3 Optimal mucosal efficacy of STm[Aux] is viability dependent. a** Germ-free mice were enterally conditioned with four doses of $10^{10}$ CFU STm[Aux] (filled red triangles, $n = 10$ animals), PAA-killed STm[Aux] (filled blue diamonds, $n = 11$ animals), or were left untreated (gray-filled circles, $n = 10$ animals). Four weeks after the first treatment (day 0) mice were challenged with wild-type STm ($10^4$ CFU). Mice were sacrificed at day 1 and day 4 after challenge with wild-type STm, respectively. Each symbol represents one individual. **b** STm-specific titer ($\log EC_{50}$) of intestinal secretory IgA determined by live bacterial flow cytometry. **c, d** Lipocalin-2 concentration in feces (**c**) and cecal content (**d**) at day 0–4 after challenge. (**e**) Cecal histopathology score at days 1 and 4 after challenge, respectively. Each symbol represents one individual. Data points depicted with an arrow are shown in panel **f**. **f** Cecal histology at day 1 and day 4 after challenge, respectively. H&E staining of cryosections. Scale bar = 100 μm. **g** Bacterial burden of wild-type STm in mLNs, spleens, and livers at days 1 and 4 after challenge, respectively. **h** Fecal colonization of wild-type STm at days 0–4 after challenge (inset graph: quantification of STm[Aux] in feces at day 0 confirming germ-free status at day 0). **i** Cecal luminal colonization of wild-type STm at day 4 after challenge. Statistics: bars indicate mean (**b–d**, **g–i**) or median values (**e**). Horizontal, dotted lines indicate the detection limit (**b–d**, **g–i**). Panel **b** was analyzed with an unpaired two-tailed $t$-test (control group excluded from test). Panels **c**, **d**, and **g** were analyzed with ordinary one-way ANOVA and Dunnett's post hoc test (comparison to control group). Panel **e** was analyzed with a two-sided multicomparisons Kruskal–Wallis test and Dunn's post hoc test. Source data are provided as a Source Data file. Detailed statistical metrics are available in the Supplementary Statistical Analysis file.

**PRR signaling redundancy in induction of immunity**. Innate pathogen recognition through PRRs is critical in the defense against primary STm infection. MYD88 knockout mice lacking TLR and IL1R family downstream signaling are consequently severely impaired in innate immunity against mucosal STm infection[17,19]. Canonical Caspase-1-dependent and non-canonical Caspase-11 (Caspase-4 in humans)-dependent inflammasome activation have also been implicated in innate immune control of STm infection[32]. The NLRC4 inflammasome is activated by the SPI1 T3SS needle complex proteins and therefore may mediate innate recognition of T3SS-competent intestinal STm specifically[3]. Moreover, the NOD1/NOD2 nodosome has been reported to respond to bacterial pathogenicity by sensing the cytoplasmic activities of *Salmonella* SPI1 T3SS-1 effector proteins[4].

Are these also factors individually important for the induction of functional adaptive immunity in the absence of an inflammatory response? To address this we tested the hypothesis that deficiencies for innate recognition pathways critical in innate immune defense also affect induction of functional adaptive immunity by live STm[Aux]. Mice deficient in (i) TLR/IL1R family adaptor proteins MYD88 and TRIF, (ii) Caspase-1 and Caspase-11, (iii) NLRC4, and (iv) NOD1 and NOD2 were derived germ free and compared with innate immunocompetent control mice for their adaptive immune responses towards live STm[Aux].

Using STm[Aux] avoids bacterial overgrowth of severely innate immunodeficient hosts that lack control and containment of intestinal microbes, leading to increased mucosal penetration also of attenuated, avirulent, and commensal bacteria. In MYD88/TRIF double-deficient mice this has been shown to result in abnormally high systemic exposure to gut commensals and consequent compensatory B cell immunity. Involvement of redundant innate signaling pathways triggered by massively increased microbial loads has been postulated to be responsible but has not been characterized further[33]. The STm[Aux] model in germ-free mice, however, uniquely fixes the bacterial load per animal and consequently avoids bacterial overgrowth to skew immune activation.

First, germ-free MYD88/TRIF double-deficient and wild-type control mice were enterally conditioned with live STm[Aux]. Four weeks after the first treatment all mice were challenged orally with wild-type STm harboring an intracellularly inducible GFP reporter plasmid (pM973)[17] and studied at day 3 after the challenge (Fig. 5a). Control groups of both genotypes were challenged but without STm[Aux] preconditioning. Quantification of cecal mucosal (Fig. 5b, Supplementary Fig. 11A), mLN, liver, and spleen (Fig. 5c) burdens of STm[pM973] revealed that the induction of functional immunity by live STm[Aux] was robust even in the highly susceptible MYD88/TRIF double-deficient mice. At day 3 post challenge, bacterial loads in mLN, livers, and

spleens of STm[Aux]-treated MYD88/TRIF-deficient mice were similar to, and in cecal mucosal tissues even lower than, those in the STm[Aux]-treated wild-type animals. In accordance with this relatively greater effect of STm[Aux] treatment in mutant than wild-type mice, a two-way ANOVA revealed a significant interaction between the effects of genotype and STm[Aux] treatment ($p < 0.0001$ for all panels; for detailed statistical metrics see Statistical Analysis file available as Supplementary Information). Quantitation of cecal luminal lipocalin-2 and histopathology scores both confirmed the protective effects of STm[Aux] treatment within each mouse genotype (Supplementary Fig. 11B, C), although these two readouts themselves are MYD88/TRIF-dependent[17,34] and should therefore be compared between both mouse genotypes with caution.

We next tested the ability of live STm[Aux] to induce adaptive immunity in NOD1/NOD2-double-knockout mice (Fig. 6a–c), NLRC4-deficient mice, and Caspase-1/11 double-deficient mice (Fig. 6d–f), all of which at day 3 of challenge were found to have no deficiency in mounting functional mucosal immunity towards live STm[Aux] conditioning (two-way ANOVA, for detailed statistical metrics see Statistical Analysis file available in the Supplementary Information).

These results show that MYD88/TRIF, Caspase-1/Caspase-11 inflammasome, and NOD1/NOD2 nodosome signaling were individually redundant for the induction of adaptive immunity by live STm[Aux] in the absence of inflammation. Their role in complementing adaptive immunity in pathogen clearance at later stages of secondary infection is likely functionally important, although not apparent at day 3 of challenge.

**Microbiota-dependent colonization and niche competition**. So far, the fully reversible germ-free mouse model uniquely had allowed the quantitative study of the immunogenicity of different phenotypes of STm[Aux] in a very clean system. However, in real-life situations STm[Aux] would interact also with the indigenous gut microbiota, which we hypothesized to provide crossfeeding of the required cell wall metabolites in vivo. This may delay STm[Aux] intestinal luminal clearance in colonized mice. We tested this hypothesis using a well-established gnotobiotic mouse model that is stably colonized with 12 representative murine intestinal taxa [stable defined moderately diverse mouse microbiota (sDMDMm)[35]] all of which are fully sequenced and openly available as pure cultures from the "Deutsche Sammlung von Mikroorganismen und Zellkulturen" (DSMZ)[36,37]. The sDMDMm model has proven merit for the study of intestinal STm infection and its interaction with the commensal microbiota without the need for harsh antibiotic treatments, and shows relevant phenotypic effects such as limiting the colonization of STm[38].

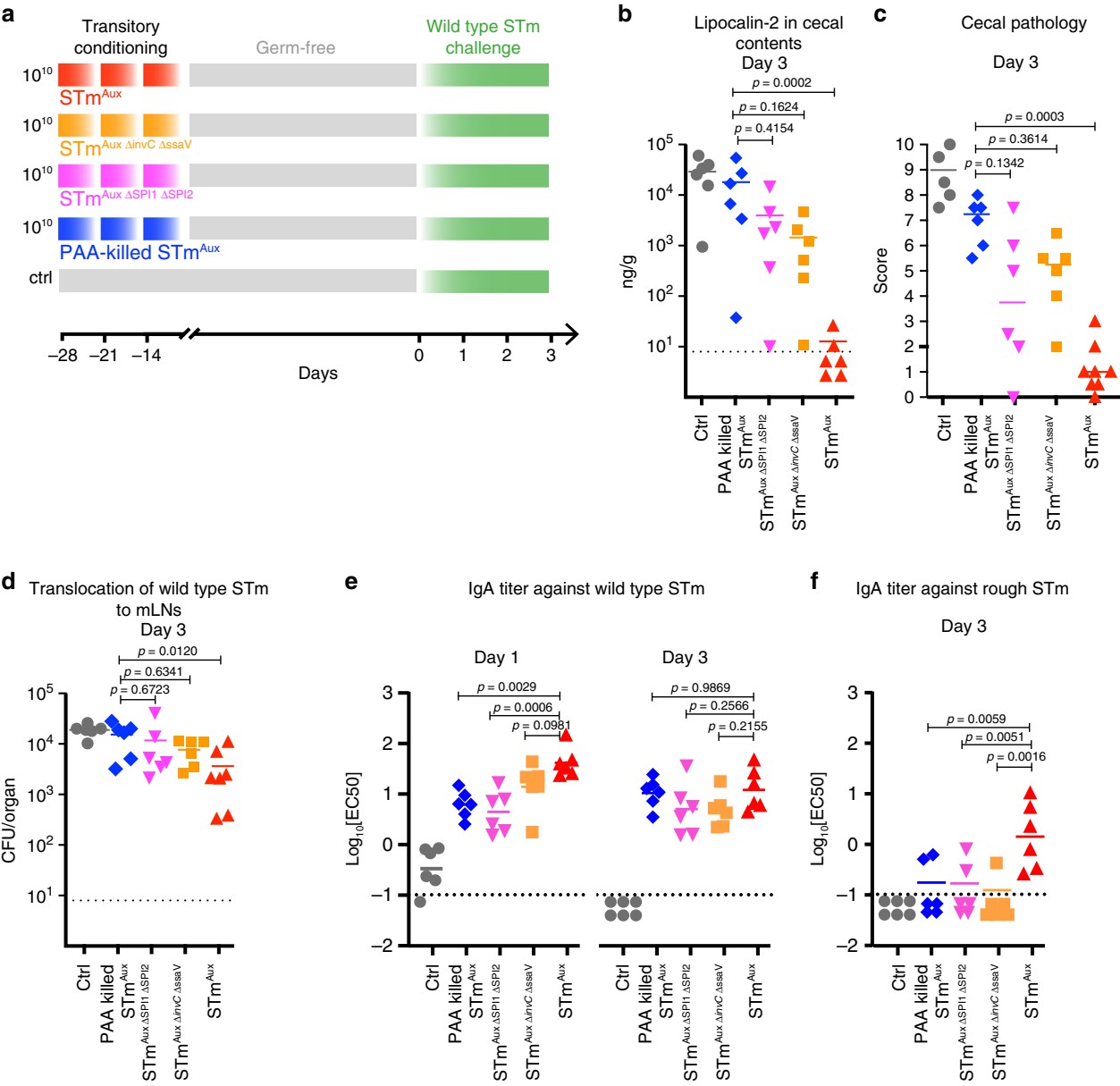

**Fig. 4 Salmonella type 3 secretion signifies robust live STm mucosal immunogenicity. a** Germ-free mice were enterally conditioned with three successive doses of $10^{10}$ CFU of live STm$^{Aux}$ (red upright triangles, $n = 6$ animals examined over two independent experiments), live T3SS-double-deficient STm$^{Aux}$ (STm$^{Aux}$ $\Delta invC$ $\Delta ssaV$, orange squares, $n = 6$ animals examined over two independent experiments), a live STm$^{Aux}$ T3SS-double-deficient SPI1/SPI2 double-deletion mutant (STm$^{Aux}$ $\Delta SPI1$ $\Delta SPI2$; purple inverted triangles, $n = 6$ animals examined over two independent experiments), PAA-killed STm$^{Aux}$ (blue diamonds, $n = 6$ animals examined over two independent experiments), or PBS vehicle only (ctrl; gray circles, $n = 6$ animals examined over two independent experiments). Four weeks after the first treatment (day 0) all mice were challenged with wild-type STm ($10^3$ CFU). Mice were analyzed on day 3 after challenge. Each symbol represents one individual. **b** Lipocalin-2 concentration in cecal contents at day 3 after challenge. **c** Cecal histopathology score at day 3 after challenge. Each symbol represents one individual. **d** Bacterial burden of wild-type STm recoverable from mLNs at day 3 after challenge. **e** STm-specific titer of intestinal secretory IgA at days 1 and 3 after challenge determined by live bacterial flow cytometry. **f** Rough STm-specific titer of intestinal secretory IgA at day 3 after challenge determined by live bacterial flow cytometry. Statistics: bars indicate mean (**b**, **d**, **e**) or median values (**c**). Horizontal, dotted lines indicate the detection limit (**b**, **d**, **e**). Panels **b**, **d**, and **e** were analyzed with ordinary one-way ANOVA and Dunnett's post hoc test (comparison to STm$^{Aux}$ group). Panel **c** was analyzed with a two-sided multicomparisons Kruskal–Wallis test and Dunn's post hoc test. The data were pooled from three independent experiments. Source data are provided as a Source Data file. Detailed statistical metrics are available in the Supplementary Statistical Analysis file.

Following a single inoculation with $10^7$ STm$^{Aux}$ by gavage, sDMDMm mice showed efficient and stable colonization of STm$^{Aux}$, reaching luminal densities similar to those of isogenic non-auxotrophic strains, including partly attenuated SPI2 TTSS-deficient ($\Delta ssaV$) and avirulent SPI1/SPI2 double-deficient ($\Delta invC$ $\Delta ssaV$) STm (Fig. 7a). STm$^{Aux}$ did not revert

to lose its auxotrophic phenotype during these experiments (no recovery of STm growth from ex vivo intestinal samples in non-supplemented control medium). Even STm$^{Aux}$ re-isolated from an sDMDMm mouse after 8 months colonized germ-free mice fully reversibly. Following gavage of $10^{10}$ CFU of either the 8-month re-isolate or the original lab strain, all mice ($n = 5$

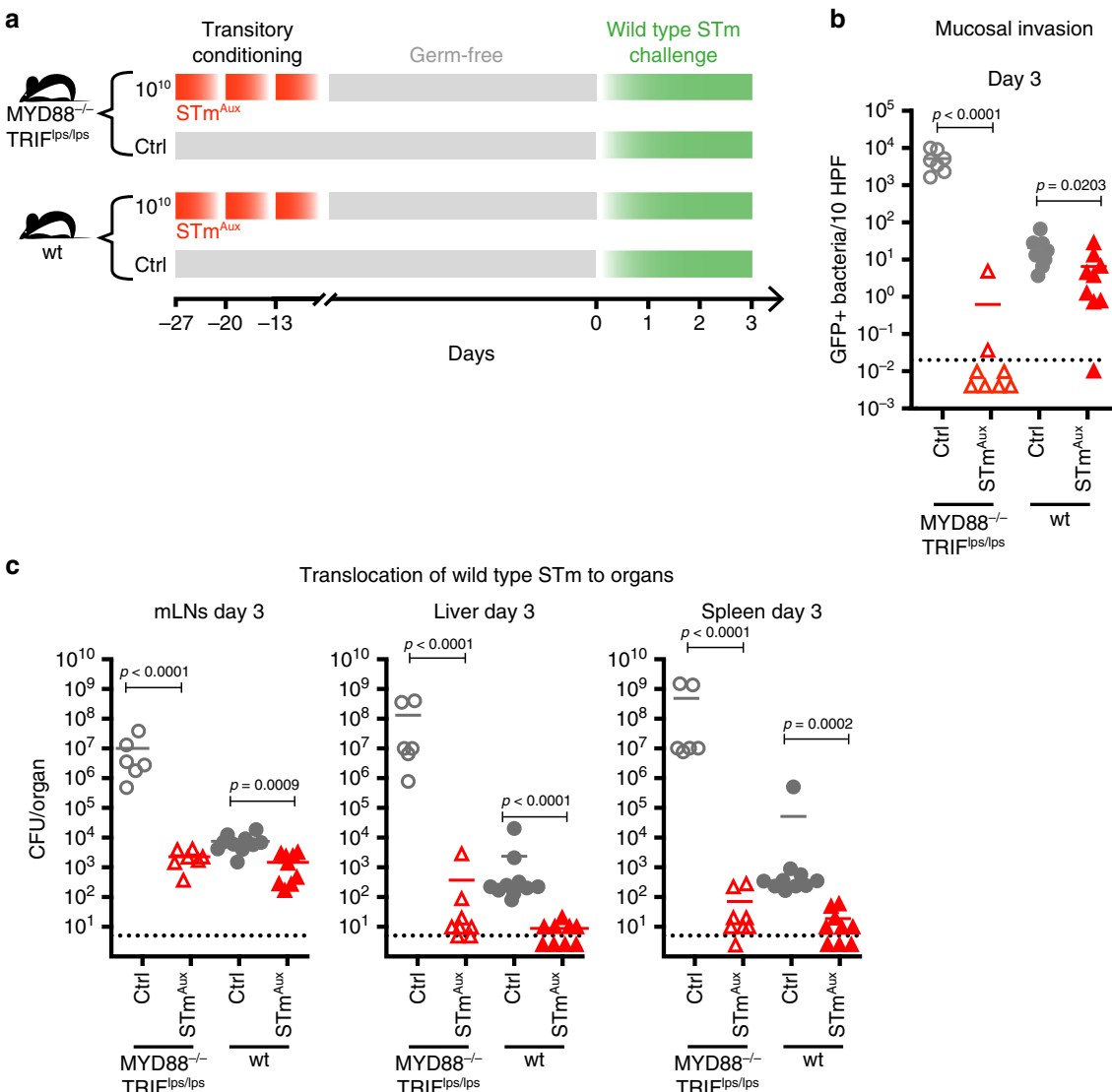

**Fig. 5 Mucosal induction of adaptive immunity by live STm^Aux is robust in MYD88/TRIF double-deficient mice. a** Germ-free MYD88^−/−TRIF^lps/lps mice (open symbols) and wild-type control mice (filled symbols) were enterally conditioned with three doses of $10^{10}$ CFU of live STm^Aux (red triangles, $n = 7$ MYD88/TRIF KO animals and $n = 9$ wild-type animals examined over two independent experiments) or left untreated as controls (gray circles, $n = 6$ MYD88/TRIF KO animals and $n = 10$ wild-type animals examined over two independent experiments). Twenty-seven days after the first treatment (day 0) mice were challenged with wild-type STm ($10^3$ CFU) harboring *ssag::eGFP* reporter plasmid pM973. The mice were studied at day 3 after challenge. Each symbol represents one individual. **b** Quantification of intracellular wild-type STm expressing eGFP in the cecal mucosa. **c** Bacterial burden of wild-type STm recoverable from mLN, spleen, and liver at day 3 after challenge. Statistics: bars indicate mean (**b, c**). Horizontal, dotted lines indicate the lower detection limit (**b, c**). Panels **b** and **c** were analyzed with two-way ANOVA (host genotype and treatment as factors) and Sidak multiple comparison correction. The data were pooled from two independent experiments. Source data are provided as a Source Data file. Detailed statistical metrics are available in the Supplementary Statistical Analysis file.

per group) had recovered to germ-free status at day 2 post inoculation.

Despite efficient luminal colonization and evidence for epithelial invasion of STm^Aux (Fig. 7e), neither deep mucosal penetration to mLN and systemic organs nor mucosal pathology were evident in either wild type (Fig. 7b–d) or MYD88/TRIF-deficient (Fig. 7f) sDMDMm mice. Thus, while crossfeeding by sDMDMm organisms can rescue gut luminal colonization, it was insufficient to recover pathogenicity of STm^Aux, which is consistent with the local intestinal luminal confinement of the crossfeeding microbiota and the activity of D-amino acid degrading enzymes in host tissues and intestinal mucus[39].

Nevertheless, induction of STm-specific IgA was seen after 4 weeks of colonization, which, in contrast to colonization

efficiency, was dependent on SPI1 T3SS competence (Fig. 7g, Supplementary Fig. 12). As an added host benefit, stably colonizing STm^Aux further provided robust niche competition to a subsequent oral challenge by wild-type STm (Fig. 7h blue symbols, and Supplementary Fig. 13B). Notably, pre-colonization with SPI1 T3SS-incompetent STm^Aux Δ*invC* provided only partial niche competition (Fig. 7h, black symbols, Supplementary Fig. 13B). SPI1/SPI2 double-deficient STm^Aux T3SS− showed the exact same phenotype (Fig. 7h, green symbols, Supplementary Fig. 13B), supporting the conclusion that SPI1 T3SS-dependent virulence factors are mainly responsible. In RAG knockout mice also T3SS-competent STm^Aux showed inefficient intestinal niche competition (Supplementary Fig. 14A–D). These findings are consistent with the interpretation that STm^Aux-induced host

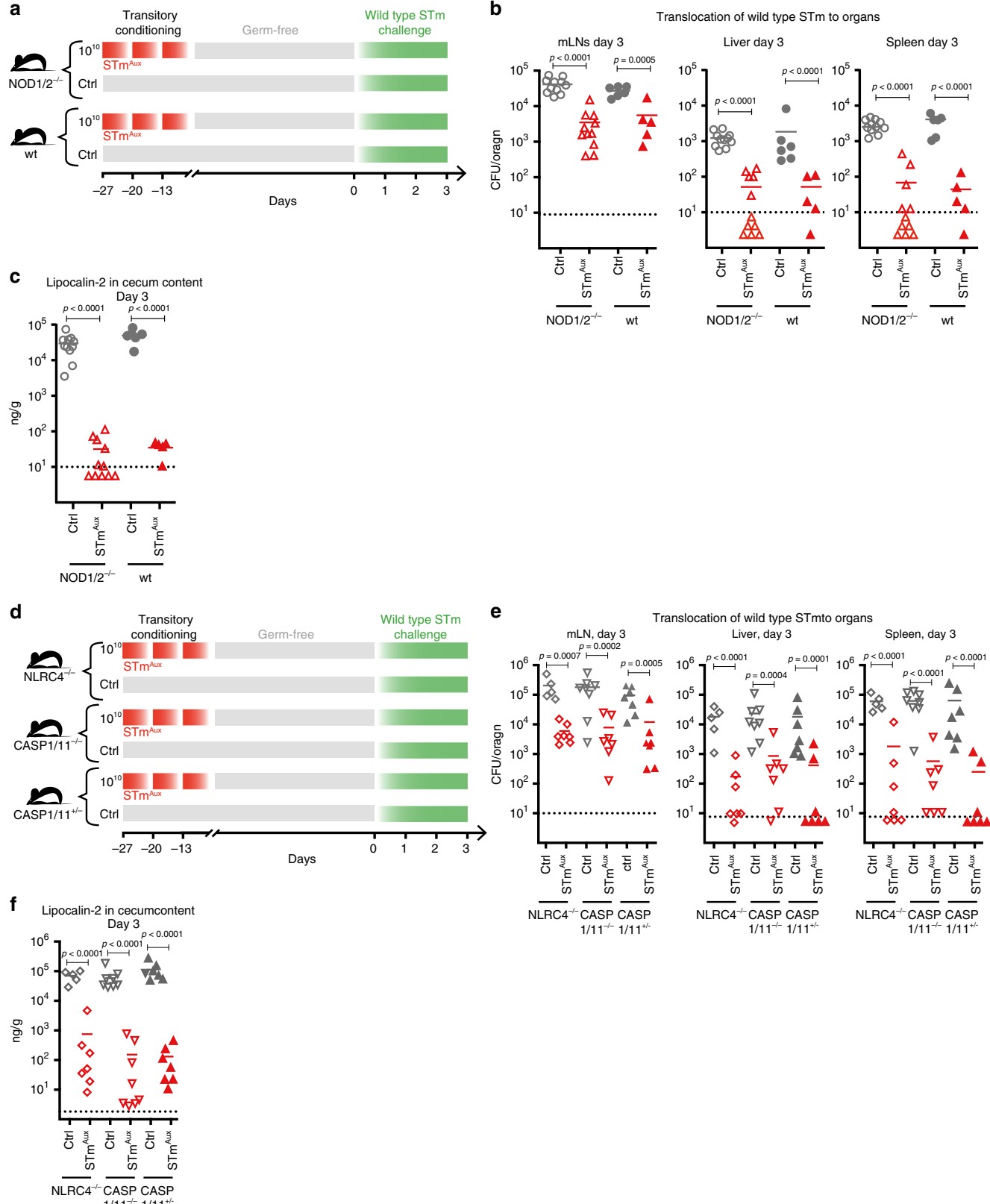

immunity synergizes with niche competition by STm[Aux] in protection against wild-type STm challenge. Measurements of cecal luminal lipocalin-2 and challenge bacterial burden in mLN, liver, and spleen at day 4 of wild-type STm challenge support this conclusion (Fig. 7i, j, Supplementary Fig. 14D). In a second context, streptomycin pre-treated conventional mice, a widely used mouse model for nontyphoid invasive salmonellosis[9,24,40], were also permissive for extended gut luminal colonization of STm[Aux] (Supplementary Fig. 13C).

These data show that in the colonized mouse model, microbiota-syntrophic STm[Aux] more closely mimics the natural pathogen in terms of intestinal luminal colonization and virulence factor-driven

**Fig. 6 NOD1/2, NLRC4 and Caspase-1/11 are individually redundant for mucosal induction of adaptive immunity by live STm^Aux. a** Germ-free NOD1/2-double-deficient mice (open symbols) and wild-type control mice (filled symbols) were either enterally conditioned with three doses of $10^{10}$ CFU of STm^Aux (red triangles, $n = 11$ NOD1/2 KO animals and $n = 5$ wild-type animals examined over two independent experiments) or left untreated (gray circles, $n = 11$ NOD1/2 KO animals and $n = 6$ wild-type animals examined over two independent experiments). Twenty-seven days after the first treatment (day 0) all mice were challenged with wild-type STm ($10^3$ CFU) and sacrificed at day 3 after challenge. **b** Bacterial burden of wild-type STm recoverable from mLN, spleen, and liver at day 3 after challenge. **c** Lipocalin-2 concentration in cecal contents at day 3 after challenge. **d** Germ-free NLRC4$^{-/-}$ mice (open diamonds), Caspase-1/11$^{-/-}$ mice (CASP1/11$^{-/-}$, open triangles), and control mice (CASP1/11$^{+/-}$ NLRC4$^{+/+}$ littermate control mice; filled triangles) were either enterally conditioned with three doses of $10^{10}$ CFU of STm^Aux (red symbols, $n = 7$ NLRC4$^{-/-}$ animals, $n = 7$ Caspase-1/11$^{-/-}$, and $n = 7$ Caspase-1/11$^{+/-}$ animals examined over two independent experiments) or left untreated (gray symbols, $n = 5$ NLRC4$^{-/-}$ animals, $n = 8$ Caspase-1/11$^{-/-}$, and $n = 7$ Caspase-1/11$^{+/-}$ animals examined over two independent experiments). Twenty-seven days after the first treatment (day 0) mice were challenged with wild-type STm ($10^3$ CFU) and sacrificed at day 3 after challenge. **e** Bacterial burden of wild-type STm recoverable from mLN, spleen, and liver at day 3 after challenge. **f** Lipocalin-2 concentration in cecal contents at day 3 after challenge. Each symbol represents one individual. Statistics: bars indicate mean (**b**, **c**, **e**, **f**). Horizontal, dotted lines indicate the lower detection limit. Panels **b**, **c**, **e**, **f** were analyzed with a two-way ANOVA (host genotype and treatment as factors) and Sidak multiple comparison correction. The data were pooled from two independent experiments. Source data are provided as a Source Data file. Detailed statistical metrics are available in the Supplementary Statistical Analysis file.

induction of IgA immunity[12]. Thus, independently of germ-free conditions, also stable intestinal STm^Aux colonization allows uncoupling of intestinal immunogenicity from pathogenicity, with the added benefit of luminal pathogen niche competition.

## Discussion

Fully attenuated or inactivated pathogens have long been noted to be poorly protective mucosal immunogens compared to more virulent strains[11–13,41]. This has been attributed mainly to the capacity of virulent pathogens to induce more vigorous innate immune responses[6] and to penetrate into and overgrow inductive sites of the mucosal immune system[11]. Here we show specifically that invasive *Salmonella* cells expressing live type 3 secretion systems are recognized by the immunogenic response, independently of their propensity to deeply penetrate and replicate as live organisms inside host tissues or the induction of a marked mucosal inflammatory response. These data suggest that the mucosal immune system reacts not only to a damaging infection but can also recognize stereotypic activities of pathogens more directly, and thus potentially more sensitively and rapidly. Consequently, a small number of highly transient mucosal exposures with virulence factor-competent STm^Aux robustly induce highly effective immunity in germ-free mice, in the absence of an inflammatory response. The underlying B cell response induced by live, virulence factor-competent STm^Aux is characterized by the production of intestinal IgA with increased O-serotype-independent *Salmonella* surface reactivity. Additional future work will be required to address which effector T cell activities may additionally contribute to STm^Aux-induced immunity[42].

It has been described previously that bacterial viability itself is an important determinant of bacterial immunogenicity, independent of pathogenicity and replication competence[43]. Live apathogenic bacteria are more immunogenic live than killed when administered parenterally[43]. This difference was revealed to be mediated by innate immune recognition of bacterial messenger RNA (mRNA), highly unstable, hence normally viability-associated, molecules. The underlying sensing pathway for bacterial mRNA was shown to be dependent on TLR8 and TRIF in humans, and on TRIF, Caspase-1 and Caspase-11 in mice[43–45]. Here we observed only a minor difference in intestinal mucosal immunogenicity between avirulent live and killed STm^Aux (see Fig. 4), which may be mediated by the same mechanism. Virulence factor-competent invasive STm^Aux, however, was much more efficacious. Its epithelial invasiveness may increase sub-epithelial live antigen delivery and consequently prime immunity more efficiently by delivering live bacteria into the tissues. However, its immunogenicity was robust even in MYD88/TRIF and Caspase-1/-11 deficient mice, and thus may not be fully explained by the same live bacterial sensing pathway. Our data confirm and extend previous findings of functional redundancy between innate and adaptive immune responses in the control of intestinal commensal bacteria[33] and the efficacy of established model vaccines with adjuvant[46] in MYD88/TRIF double-deficient mice. Here we show that this extends also to intestinal pathogenic bacteria. The remarkable robustness of this system may represent an evolutionary adaptation to pathogens that evade or alter the innate immune defense.

Long-established live STm vaccine strains like SPI2 T3SS-deficient[47] and aromatic amino acid auxotrophic *aro* mutants[48] of STm also are effective mucosal immunogens, but are not fully growth deficient in host tissues and consequently considered dangerous for HIV positive and other immunocompromised individuals (reviewed in ref. [14]). This has so far ruled out approval for human application. On the other hand, peptidoglycan metabolite auxotrophic STm strains similar to the one we presented in this paper have been developed previously[49] but in this form have been considered insufficiently immunogenic because of their poor mucosal penetration. This conclusion is however predicated on the preclinical study mainly in conventional rodent models that are (like humans) intestinally colonization resistant against *Salmonella*[9]. In this context, when STm^Aux proliferation in the intestinal lumen is inhibited by the competing microbiota, its colonization dynamics would be expected to be more similar to the germ-free mouse model, and it may consequently require very high oral doses (as we saw in germ-free mice) to be efficacious. Instead, the field has moved into the direction of developing more sophisticated strains that display regulated delayed in vivo attenuation/lethality phenotypes, allowing for transient survival, replication, and tissue invasion in vivo[11]. These highly innovative approaches are inherently more difficult to combine with safety parameters matching those of the constitutively D-Ala/Dap auxotrophic strain. The presented experiments in non-colonization-resistant mouse models highlight yet another possible strategy. The remarkably efficient gut luminal microbiota-syntrophy permitted extended mucosal stimulation with live virulence factor-competent STm^Aux, without compromising the strain's deficiency in causing pathology and systemic infection. This phenotype could potentially be exploited further by metabolic engineering of STm^Aux strains to gain intestinal colonization efficiency, or by temporal reduction of colonization resistance in the host at the time of treatment (preferably other than by antibiotic treatment). However, given that our conclusions so far are based on mouse models that have laboratory levels of microbiota complexity, additional work in more relevant preclinical models will be necessary to assess potential translatability of these findings for veterinary or human medical applications.

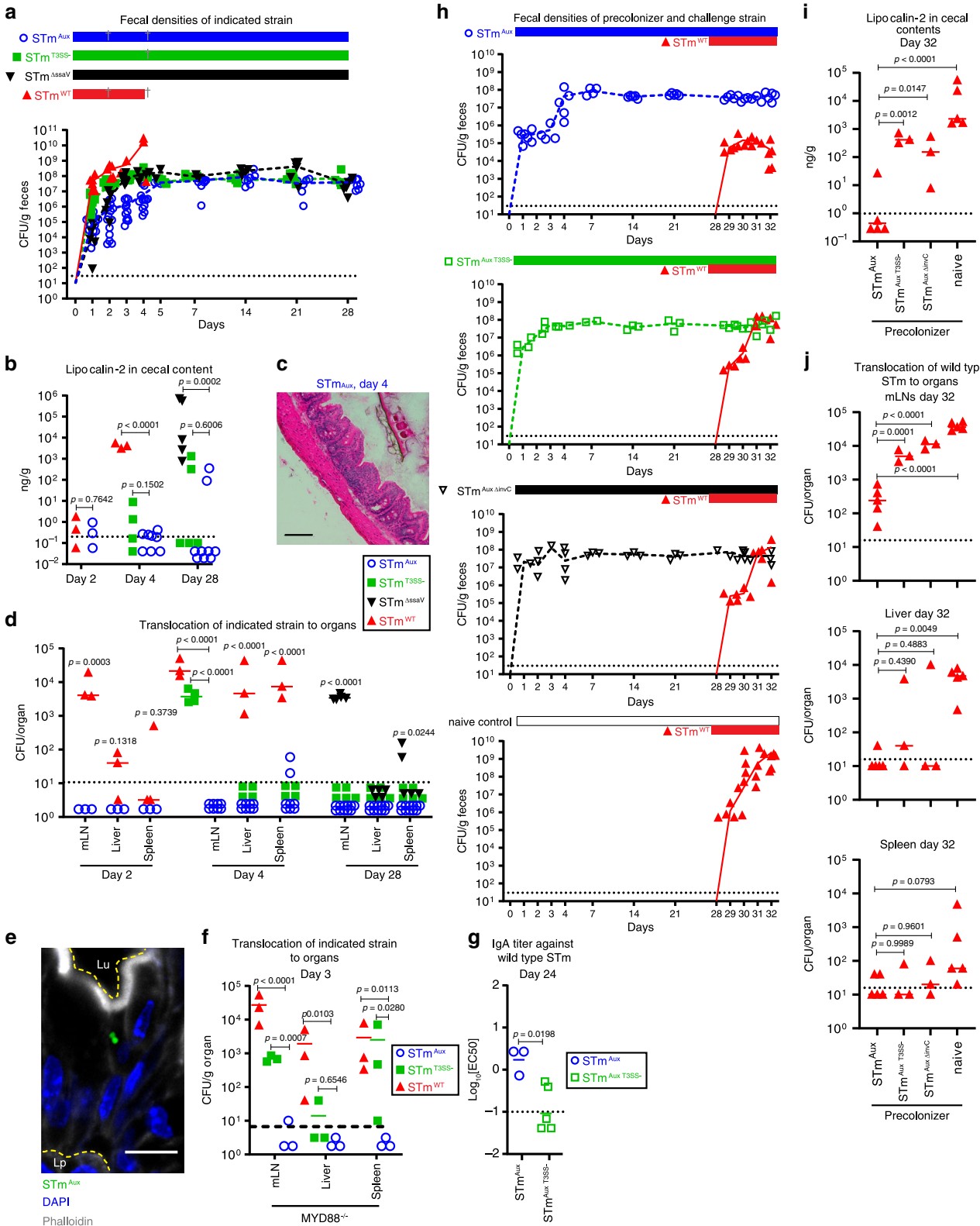

## Methods

**Bacterial strains, plasmids, and culture media**. The bacterial strains and plasmids used in this study are listed in Table 1. Auxotrophic strain HA135 (STm^Aux, UK-1 background) [ΔmetC::TetRA Δalr ΔdadX Δasd] was generated from strain χ9052 [Δalr3 ΔdadB4 ΔasdA33] by replacing the coding region of metC with a TetRA resistance cassette by Lambda Red recombineering using recombineering plasmid pSIM5 (ref. [50]) as described in ref. [51]. Isogenic mutant alleles ΔinvC::aphT, ΔinvC::aphT ΔssaV::cat, and Δ(avrA-invH::cat) Δ(ssaG-ssaU::aphT) were

transferred into the STm and STm^Aux backgrounds by phage P22-mediated transduction using the donor strains M736, M73831 and χ9650 (ref. [28]), respectively, as described[52]. Auxotrophic strain HA623 (SL1344 background) [ΔmetC ΔalrN ΔalrP Δasd] was generated from strain SL1344 (SB300) by in-frame deletion of each gene. This was achieved be generation of four single deletion mutants in SL1344 using the plasmid pSIM6 encoded Lambda Red recombinase system[50] for allelic exchange of the coding sequence (leaving the stop codon) with a Tet selectable tetA-sacB cassette, followed by four sequential rounds of P22

**Fig. 7 Efficient colonization and immune induction by auxotrophic STm in microbiota-associated mice. a** sDMDMm mice were gavaged with a single dose of $10^7$ CFU of either STm$^{Aux}$ (blue open circles, $n = 21$), STm$^{T3SS-}$ (green squares, $n = 9$), STm$^{\Delta ssaV}$ (black triangles, $n = 5$), or wild-type STm (red triangles, $n = 6$). Time course of viable bacteria of each strain recoverable from feces. **b** Lipocalin-2 concentration in cecal contents at day 2, 4, and 28 after inoculation with the indicated STm strains. Pictures show representative H&E stainings of ceca at day 4 post inoculation with either STm$^{WT}$ or STm$^{Aux}$. **c** Representative cecal histology on day 4 after colonization with STm$^{Aux}$, H&E staining, scale bar 100 µm. **d** Bacterial burden of indicated STm strains recoverable from mLN, spleen, and liver at day 2, 4, and 28 after initial colonization. **e** Confocal immunofluorescence microscopy of cecum tissue showing an epithelial cell invaded by STm$^{Aux}$. Green, STm$^{Aux}$ harboring *ssag::eGFP* reporter plasmid pM973; blue, DNA (DAPI); gray, F-actin (phalloidin). Dotted yellow lines outline the border of the epithelium facing the intestinal lumen (Lu) and lamina propria (Lp), respectively. Scale bar = 10 µm ($n = 6$ animals). **f** Bacterial burden of indicated STm strains recoverable from mLN, liver, and spleen at day 3 post inoculation in MYD88/TRIF double KO mice ($n = 3$ animals per treatment group). **g** Wild-type STm-specific titer of intestinal IgA at day 24 post inoculation of sDMDMm mice with either STm$^{Aux}$ (blue open circles, $n = 3$ animals) or STm$^{Aux\ T3SS-}$ (green open squares, $n = 5$ animals). **h** sDMDMm mice were inoculated by gavage with a single dose of $10^7$ CFU of either STm$^{Aux}$ (blue open circles, $n = 5$ animals), STm$^{Aux\ T3SS-}$ (green open squares, $n = 3$ animals), STm$^{Aux\ \Delta ssaV}$ (black open triangles, $n = 4$ animals), or left untreated ($n = 5$ animals), and at day 28 challenged with $10^7$ CFU wild-type STm (red open triangles). Mice were studied 4 days after challenge (day 32). **i** Lipocalin-2 concentration in cecal contents quantified at day 32 (4 days after challenge). **j** Bacterial burden of wild-type STm recoverable from mLN, liver, and spleen at day 32. Each symbol represents on individual. Statistics: connecting lines connect means (**a**, **f**), bars indicate mean (**b–e**, **g**, **h**). Horizontal dotted lines indicate the lower detection limit (**a–h**). Panels **b–d**, **g**, **h** were analyzed with a one-way ANOVA and Dunnett's post hoc test or unpaired two-tailed *t*-test (**b**, **c**, **e**). Data shown in **a–d** are pooled from four independent internally controlled experiments. Data shown in **h–j** were obtained from the same experiment. Source data are provided as a Source Data file. Detailed statistical metrics are reported in the Supplementary Statistical Analysis file.

### Table 1 Bacterial strains and plasmids.

| Strain (acronym)/plasmid | Genetic background | Relevant genotype | Known resistances | Comments | Origin or reference |
|---|---|---|---|---|---|
| χ4138 (STm) | UK-1 | Wild-type strain | Nal | Virulent wild-type control | 28 |
| χ9650 (STm$^{\Delta SPI1\ \Delta SPI2}$) | UK-1 | Δ(*avrA-invH*::cat) Δ(*ssaG-ssaU*::aphT) | Nal, Cam, Kan | | 28 |
| χ9052 | UK-1 | Δ*alr3* Δ*dadB4* Δ*asdA33* | | | 68 |
| M736 | ATCC 14028s derivate IR715 (ref. 69) | Δ*invC*::aphT | Nal, Kan | | 31 |
| M738 | ATCC 14028s derivate IR715 (ref. 69) | Δ*ssaV*::cat | Nal, Cam | | 31 |
| HA135 (STm$^{Aux}$) | UK-1, χ9052 | Δ*metC*::tetRA Δ*alr3* Δ*dadB4* Δ*asdA33* | Tet | | This study |
| HA618 (STm$^{Aux\ \Delta invC}$) | UK-1, STm$^{Aux}$ | Δ*metC*::tetRA Δ*alr3* Δ*dadB4* Δ*asdA33* Δ*invC*::aphT | Tet, Kan | | This study |
| HA218 (STm$^{Aux\ \Delta invC\ \Delta ssaV}$) | UK-1, STm$^{Aux}$ | Δ*metC*::tetRA Δ*alr3* Δ*dadB4* Δ*asdA33* Δ*invC*::aphT Δ*ssaV*::cat | Tet, Kan, Cam | | This study |
| HA208 (STm$^{Aux\ \Delta SPI1\ \Delta SPI2}$) | UK-1 | Δ*metC*::tetRA Δ*alr3* Δ*dadB4* Δ*asdA33* Δ(*avrA-invH*::cat) Δ(*ssaG-ssaU*::aphT) | Tet, Kan, Cam | | This study |
| pM973 | pWKS30 (ref. 70) | | Amp | eGFP under control of the *ssaG* promoter | 17 |
| SB300 (STm) | SL1344 | Wild-type strain | Str | | |
| HA623 | SL1344 | Δ*metC* Δ*alrN* Δ*alrP* Δ*asd* | Str | | This study |
| HA630 (STm$^{Aux}$) | SL1344, HA300 | Δ*metC* Δ*alrN* Δ*alrP* Δ*asd*::tetRA | Str, Tet | Used in Fig. 7 and corresponding Supplementary figures | This study |
| HA705 (STm$^{T3SS-}$) | SL1344, SB300 | Δ*invC*::aphT Δ*ssaV*::cat | Str, Kan, Cam | Used in Fig. 7 | This study |
| HA706 (STm$^{\Delta ssaV}$) | SL1344, SB300 | Δ*ssaV*::cat | Str, Cam | Used in Fig. 7 | This study |
| HA702 (STm$^{Aux\ T3SS-}$) | SL1344, SB300 | Δ*metC* Δ*alrN* Δ*alrP* Δ*asd*::tetRA Δ*invC*::aphT Δ*ssaV*::cat | Str, Tet, Kan, Cam | Used in Fig. 7 | This study |
| HA700 (STm$^{Aux\ \Delta invC}$) | SL1344, SB300 | Δ*metC* Δ*alrN* Δ*alrP* Δ*asd*::tetRA Δ*invC*::aphT Δ*invC*::aphT | Str, Tet, Kan | Used in Fig. 7 | This study |
| HA727 (STm$^{Aux\ \Delta Triple-Eff}$) | SL1344, SB300 | Δ*metC* Δ*alrN* Δ*alrP* Δ*asd* Δ*invC*::aphT Δ*invC*::aphT Δ*sopE*::pGP704 Δ*sipA*::aphT Δ*sopE2*::pM218 | Str, Tet, Kan, Cam, | Used in Fig. S7 | This study |
| HA733 (STm$^{\Delta Triple-Eff}$) | SL1344, SB300 | Δ*metC* Δ*alrP* Δ*sopE*::pGP70 Δ*sipA*::aphT Δ*sopE2*::pM218 | Str, Tet, Kan, Cam | Prototrophic derivative of HA727 | This study |
| SKI12 (rough STm) | SL1344, SB300 | Δ*wbaP* | Nal | | 71 |
| M1525 (SEn) | *S. Enteritidis* 125109 | Wild-type strain | | | 72 |
| HA627 (rough SEn) | *S. Enteritidis* 125109 | Δ*rfbS* | | | This study |

transduction followed by Lambda Red recombineering mediated removal of the *tetA-sacB* cassette by counterselection as described[53], leading to quadruple deletion mutant HA623. HA630 was generated by Lambda Red recombination of a *tetRA* resistance cassette into the *asd* deletion site of HA623. The mutagenesis primers used are listed in Table 2. Auxotrophic SPI1 effector gene *sopE sopE2 sipA* triple mutant H727 was constructed in parent strain SL1344 as described previously[31]. The reported avirulence phenotype of the *sopE sopE2 sipA* mutation was confirmed by gentamicin protection assay (see Fig. S7D) and by P22 transduction of the wild-type alleles of *asd* and *alrN* to recover prototropy in resultant strain HA733, which was then confirmed to be of strongly reduced intestinal virulence in germ-

free mice (histopathological score at day 2 of infection = 2.5 ± 0.5 [mean ± range; $n = 2$]; wild-type control = 11).

Luria-Bertani (LB) medium (Sigma-Aldrich) was used as standard bacterial culture medium. Ampicillin (Sigma; 100 µg/mL), tetracycline (Sigma; 12.5 µg/mL), kanamycin (Sigma; 50 µg/mL), chloramphenicol (Sigma; 6 µg/mL), nalidixic-acid (Sigma; 50 µg/mL), *meso*-diaminopimelic acid (m-Dap; Sigma, 50 µg/mL), and/or D-alanine (D-Ala; Sigma, 200 µg/mL) were added to the medium as appropriate.

**Cellular invasion assays.** HeLa (Kyoto) cells were seeded into 24-well dishes and were grown for 1 day until 80% confluence was obtained. HeLa cells were cultured

**Table 2 Primers used for bacterial mutagenesis.**

| Primer name | 5′ → 3′ sequence | Additional information | Applicable background |
|---|---|---|---|
| STM-metC-tetRA-F | TTTGGCAAAATTTTCATCTGTATCACACGTCGCCAGGGTGCAGATGGTTATATTCATGCTAGTTTAGACATCCAGACGGTTAAAATCAGGAAACGCAACTTAAGACCCACTTTCACATT | Allelic exchange of metC with TetRA | UK-1 |
| STM-metC-tatRA-R | GTACTCCTGAATCGTCCGGGATGCCTTGATCCCGGACGCAACAAACGCAGACTTTTCCACGGAAATTGTCTGCATATATGTCCATCCCCGGCAACTTTACTAAGCACTTGTCTCCTG | Allelic exchange of metC with TetRA | UK-1 |
| STM-metC-cntr-F | ACGCCAGAATCAAACCAATC | metC control primer | UK-1 |
| STM-metC-cntr-R | ATCGCCAGGTAAGAATGACG | metC control primer | UK-1 |
| STM-asd-TetA-SacB-F | GAACCACACGCAGGCCCGATAAGCGCTGCAATAGCCACTAATCAAAGGGAAAACTGTCCATATGC | Allelic exchange of asd with TetA-SacB | SL1344 |
| STM-asd-TetA-SacB-R | CGCGCATACACAGCACATCTCTTTGCAGGAAAAAAACGCTTCCTAATTTTTGTTGACACTCTATC | Allelic exchange of asd with TetA-SacB | SL1344 |
| STM-asd-rmvl-F | GAACCACACGCAGGCCCGATAAGCGCTGCAATAGCCACTAAGCGTTTTTTCCTGCAAAGAGATGTGCTGTGTATGCGCG | Removal of TetA-SacB cassette | SL1344 |
| STM-asd-rmvl-R | CGCGCATACACAGCACATCTCTTTGCAGGAAAAAAACGCTTAGTGGCTATTGCAGCGCTTATCGGGCCTGCGTGTGGTTC | Removal of TetA-SacB cassette | SL1344 |
| STM-asd-ctrl-Rv | TAAGCGCTGCAATAGCCACT | asd control primer | SL1344 |
| STM-asd-ctrl-Fw | TTGCGACTTTGGCTGCTTTT | asd control primer | SL1344 |
| STM-alrN-TetA-SacB-F | CCCAAGTGGACCGGTCGACGCCTTAGCCTGAATTAGGTTAATCAAAGGGAAAACTGTCCATATGC | Allelic exchange of alrN with TetA-SacB | SL1344 |
| STM-alrN-TetA-SacB-R | CAACGTTTGCATAGCGCGCATAACTGATAAAGGAAGTGAATCCTAATTTTTGTTGACACTCTATC | Allelic exchange of alrN with TetA-SacB | SL1344 |
| STM-alrN-rmvl-F | CCCAAGTGGACCGGTCGACGCCTTAGCCTGAATTAGGTTATTCACTTCCTTTATCAGTTATGCGCGCTATGCAAACGTTG | Removal of TetA-SacB cassette | SL1344 |
| STM-alrN-rmvl-R | CAACGTTTGCATAGCGCGCATAACTGATAAAGGAAGTGAATAACCTAATTCAGGCTAAGGCGTCGACCGGTCCACTTGGG | Removal of TetA-SacB cassette | SL1344 |
| STM-alrN-ctrl-Fw | GTTTGGCGGCATGATTTGGA | alrN control primer | SL1344 |
| STM-alrN-ctrl-Rv | CACCTTAGGCTGGACGATGG | alrN control primer | SL1344 |
| STM-alrP-TetA-SacB-F | GATGAGTAACTCTCCGTCATTCTTTTAACAAGGAATTCAATCCTAATTTTTGTTGACACTCTATC | Allelic exchange of alrP with TetA-SacB | SL1344 |
| STM-alrP-TetA-SacB-R | CCGGATAAGCGCAAGCGCCACCCGGCCCGCCGCGTATTTAATCAAAGGGAAAACTGTCCATATGC | Allelic exchange of alrP with TetA-SacB | SL1344 |
| STM-alrP-rmvl-F | GATGAGTAACTCTCCGTCATTCTTTTAACAAGGAATTCAATAAATACGCGGCGGGCCGGGTGGCGCTTGCGCTTATCCGG | Removal of TetA-SacB cassette | SL1344 |
| STM-alrP-rmvl-R | CCGGATAAGCGCAAGCGCCACCCGGCCCGCCGCGTATTTATTGAATTCCTTGTTAAAAGAATGACGGAGAGTTACTCATC | Removal of TetA-SacB cassette | SL1344 |
| STM-alrP-ctrl-Fw | GGTACGGTTCGTCTGACGTT | alrP control primer | SL1344 |
| STM-alrP-ctrl-Rv | TATTACCGGATGACGGCGTG | alrP control primer | SL1344 |
| STM-metC-TetA-SacB-F | TAGTTTAGACATCCAGACGGTTAAAATCAGGAAACGCAACTCCTAATTTTTGTTGACACTCTATC | Allelic exchange of metC with TetA-SacB | SL1344 |
| STM-metC-TetA-SacB-R | CGGAAATTGTCTGCATATATGTCCATCCCCGGCAACTTTAATCAAAGGGAAAACTGTCCATATGC | Allelic exchange of metC with TetA-SacB | SL1344 |
| STM-metC-rmvl-F | TAGTTTAGACATCCAGACGGTTAAAATCAGGAAACGCAACTAAAGTTGCCGGGGATGGACATATATGCAGACAATTTCCG | Removal of TetA-SacB cassette | SL1344 |
| STM-metC-rmvl-R | CGGAAATTGTCTGCATATATGTCCATCCCCGGCAACTTTAGTTGCGTTTCCTGATTTTAACCGTCTGGATGTCTAAACTA | Removal of TetA-SacB cassette | SL1344 |
| STM-metC-ctrl-Fw | GCCAGGGTGCAGATGGTTAT | metC control primer | SL1344 |
| STM-metC-ctrl-Rv | GACGCAACAAACGCAGACTT | metC control primer | SL1344 |
| STM-asd-TetRA-Fw | GCGCGCATACACAGCACATCTCTTTGCAGGAAAAAAACGCTTTAAGACCCACTTTCACATT | Allelic exchange of asd with TetRA | SL1344 |
| STM-asd-TetRA-Rv | AACCACACGCAGGCCCGATAAGCGCTGCAATAGCCACTACTAAGCACTTGTCTCCTG | Allelic exchange of asd with TetRA | SL1344 |

in Dulbecco modified Eagle's medium (DMEM) containing 10% fetal bovine serum (FBS) and incubated at 37 °C under an atmosphere containing 5% $CO_2$. Shortly before STm infection, the adherent cells were incubated in Hanks' buffered salt (HBSS) medium. STm strains were inoculated from a single colony in 10 mL D-Ala (200 μg/mL) and m-Dap (50 μg/mL) supplemented (optional) 0.3 M sodium-chloride/LB and incubated at 150 r.p.m., at 37 °C for 16 h. STm cultures were diluted 1:20 into 40 mL fresh medium and incubated at the same conditions for 5 h. Subsequently, STm were washed in PBS and HeLa cells were infected with approximately $3 \times 10^6$ CFU STm per well. Fifty minutes after infection, extra-cellular STm were inactivated by replacing HBSS with DMEM containing 10% FBS and gentamicin (400 μg/mL) up to a total infection time of 2 h. For the quantifi-cation of intracellular (=gentamicin protected) STm, HeLa cells were washed in PBS and subsequently lysed in 0.1% sodium-deoxycholate/PBS. The released intracellular bacteria were quantified on D-Ala and m-Dap supplemented LB agar plates.

For differential fluorescent staining of extra- and intracellular STm, HeLa cells were cultured on glass coverslips and infected as described above. After a total infection time of 2 h, cells were fixed in 4% PBS-buffered paraformaldehyde. After incubation in blocking buffer (2% BSA/PBS), the cells were incubated in rabbit-anti-STm O-antigen group B antiserum (Becton Dickinson) diluted in blocking buffer, washed twice in blocking buffer, and incubated in goat anti-rabbit CY3 (Jackson Innoreseach) diluted in blocking buffer. Subsequently, cells were permeabilized with 0.5% Triton X-100 in PBS, incubated again in blocking buffer, and stained again with rabbit-anti-STm O-antigen group B antiserum (Becton

Dickinson), washed in blocking buffer, and incubated in a solution of goat anti-rabbit Alexa Fluor-647 (Jackson Immunoresearch) antibody and DAPI (Sigma). The coverslips were mounted under Vectashield mounting medium (Vectorlabs) and examined under a Zeiss Axio Imager M1 fluorescence microscope with a ×63 oil objective and recorded with an AxioCamHR3 camera.

**Animal experiments**. Animal experiments were performed in accordance to animal experiment licenses (BE94/11, BE91/14, BE36/15, BE85/17) approved by the Bernese Cantonal Ethical committee for animal experiments and carried out in accordance with Swiss Federal law for animal experimentation.

Mice were maintained under axenic barrier conditions at the Clean Mouse Facility of the University of Bern. The housing conditions in the whole facility are strictly controlled. The ambient temperature is 23–25 °C and the relative humidity is 52–60%. All mice received at all times standard diet (Kliba 3307) and water ad libitum. The genetic background of all mice used was C57BL/6. $MYD88^{-/-}$ $TRIF^{lps/lps}$ mice[54,55] were provided by Bruce A. Beutler, The Scripps Research Institute, CA, USA and maintained germ free in the Clean Mouse Facility, University of Bern. $NRLC4^{-/-}$ (also known as $Ipaf^{-/-}$) mice[56] were provided by V. Dixit, Genentech, and derived germ free in the Clean Mouse Facility, University of Bern. $NOD1^{-/-} NOD2^{-/-}$ mice[57] were provided by Dana Philpott, University of Toronto ON, Canada, and derived germ free and provided by Elena F. Verdue, Axenic Gnotobiotic Facility, McMaster University, Hamilton ON, Canada. Caspase-1/11$^{-/-}$ mice (B6N.129S2-Casp1$^{tm1Flv}$ Casp4$^{del}$/J[58]) were purchased from

The Jackson Laboratory in the form of cryopreserved embryos and transferred into germ-free recipients in the Clean Mouse Facility, University of Bern. Germ-free and sDMDMm wild-type C57BL/6 animals, germ-free JH[−/−] mice[59] and RAG[−/−] mice[60] were maintained at the Clean Mouse Facility, University of Bern.

Gnotobiotic sDMDMm mice have been generated at the Clean Mouse Facility of the University of Bern by inoculation of germ-free C54BL/6 mice with purified culture of the murine intestinal bacterial consortium Oligo-MM12 (ref. [38]) and stably maintained in flexible film isolators under strictly axenic conditions. sDMDMm RAG[−/−] mice and MYD88[−/−] TRIF[lps/lps] mice were generated by co-housing of the genetically modified germ-free mice with gnotobiotic wild-type sDMDMm mice for 4 weeks.

SPF C57BL/6 mice were purchased from Charles River (France) and maintained at the Central Animal Facility of the Department of Biomedical Research, University of Bern. For infections in the streptomycin pretreatment model[24] SPF mice were pre-treated with 20 mg of streptomycin dissolved in sterile water prior to infection with STm by gavage.

Infection and colonization experiments were performed under strict aseptic conditions. Mice were derived and maintained germ free in flexible film isolators[61] (including the duration of transient auxotrophic bacterial conditioning) or autoclaved Sealsafe-plus IVC cages (Tecniplast, Italy; during STm challenge and short-term infections) at the Clean Mouse Facility (CMF) of the University of Bern. Animals were provided with autoclaved mouse chow (Kliba 3307) and water ad libitum. Germ-free status of all animals was routinely monitored using culture-based and culture-independent methods established by the Clean Mouse Facility, DKF University of Bern. Mice were infected with 200 μL STm suspension.

Bacteria for enteral inoculation were grown under SPI1-inducing conditions. Auxotrophic STm were inoculated into 10 mL D-Ala- (200 μg/mL) and m-Dap-(50 μg/mL) supplemented LB containing 0.3 M NaCl and incubated shaking at 150 r.p.m., at 37 °C for 16 h. The resulting bacterial cultures were diluted $10^8$-fold in 500 mL fresh medium and incubated under the same conditions for 15 more hours. STm were harvested by centrifugation (15 min, $4816 \times g$, 4 °C), washed twice with cold PBS, and resuspended to the appropriate densities. For peracetic acid (PAA) inactivation, an aliquot of auxotrophic STm was resuspended in 10 mL 1% peracetic acid for 1 h at room temperature. The inactivated STm suspension was washed with PBS and resuspended to the the appropriate density. Sterility of PAA-killed inocula was confirmed by standard culture methods. Wild-type STm cultures were inoculated from a single colony in 10 mL 0.3 M sodium-chloride/LB and incubated at 150 r.p.m., at 37 °C for 16 h. Wild-type STm cultures were diluted 1:20 into 40 mL fresh medium and incubated at the same conditions for 5 h.

**Bacterial loads in organs and feces**. Organs, feces, and cecum content were aseptically collected. Organs were homogenized in 0.5 mL 0.5% tergitol/PBS, feces and cecum content in 0.5 mL PBS using a tissue lyzer (TissueLyzer LT, Qiagen, 50 Hz, 3–5 min, with a stain-less steel bead). Bacterial loads were quantified by plating on LB agar. Where necessary, D-Ala (200 μg/mL) and m-Dap (50 μg/mL) were added.

**Isolation of intestinal secretory IgA**. Intestinal IgA lavages were collected by rinsing the small intestine with 5 mL of 1% soybean-trypsin inhibitor/0.05 M EDTA/PBS. The intestinal lavages were spun at $4816 \times g$, >20 min, 4 °C. The supernatant was sterile-filtered (0.22 μm cut-off size) to remove bacteria-sized particles and stored long-term in aliquots frozen at −20 °C.

**Immunoglobulin repertoire sequencing**. Germ-free C57BL/6 mice were orally administrated with $3 \times 10^{10}$ STm$^{Aux}$ or STm$^{Aux \, \Delta invC \, \Delta ssaV}$ three times at 7-day intervals. Naïve germ-free mice served as naïve controls. Twenty-eight days post last administration, ileum and MLN were dissected and snap-frozen in Trizol reagent (Life Technologies) using liquid nitrogen. Thawed tissues were homogenized (Retsch bead-beater) in 1 mL of Trizol reagent. Two hundred microliters chloroform was added to samples and centrifugation ($12,000 \times g$, 15 min, 4 °C) was performed. The upper phase containing RNA was collected, and RNA was precipitated with ice-cold isopropanol. After washing once with 75% (v/v) ethanol, RNA pellet was dried and resuspended in RNase-free water. Nanodrop2000 (Thermo Scientific) was used to quantify RNA concentrations and purity.

To prepare IgA amplicons, cDNA was synthesized by mixing 700 ng of RNA, 1 μL of 2 μM IgA gene specific primer mix (as previously described[62], 1 μL of 10 mM dNTP (containing dATP, dCTP, dGTP, and dTTP at a final concentration of 10 mM, Invitrogen) and topped with dH$_2$O to 13 μL. Samples were heated to 65 °C for 5 min, and then cooled for 1 min on ice. Four microliters 5X first strand buffer (Invitrogen), 1 μL 782 0.1 M DTT (Invitrogen), 1 μL RNaseOUT (Invitrogen), and 1 μL Superscript III RT enzyme (Invitrogen) were added to each reaction, mixed, and incubated at 55 °C for 50 min. A heat inactivation at 70 °C for 15 min was done to stop the reaction. Five microliters of synthesized cDNA library was used as a template DNA for amplicon PCR PlatinumTaq PCR buffer (Qiagen) following the manufacturer's instruction. Primers used in the PCR reaction listed below were described previously[63]. PCR products were electrophorized on 1.5% agarose gel and purified with the QIAquick Gel Extraction kit (Qiagen). The purified DNA was quantified using Qbit (Thermofisher). Sequencing adapters (Nextera® XT Index Kit, Illumina) were linked to each amplicon by doing a second PCR. After testing with a Fragment Analyzer™ (Advanced Analytical), amplicons with sequencing adaptors were pooled for sequencing on the MiSeq Illumina sequencer using paired 250 bp mode.

For primer sequences used view Table 3.

**Antibody repertoire sequencing analysis**. B cell IgA receptor heavy chain libraries were prepared as previously described[64] and sequenced on the Illumina MiSeq platform ($2 \times 250$ cycles, paired-end). Output files were preprocessed (VDJ alignment, clonotyping) using MiXCR (v3.0.12). Clonotypes were defined by 100% amino acid sequence identity of CDR3 regions. Annotation of the different segments was defined by MiXCR according to the nomenclature of the Immunogenetics database (IMGT)[65]. MIXCR output files were further processed in the post-processing tool-suite: VDJtools[66]. Further filtering was applied in order to keep only productive sequences if: (i) they were composed of at least four amino acids and (ii) had a minimal read count of 2 (ref. [63]) and were in-frame.

## Table 3 Primers used for antibody repertoire sequencing.

| IgH Forward Mix | 5′ → 3′ sequence Illumina Adapter sequence read 1 + Diversity region + VH 5′-specific region |
|---|---|
| IgH-UAd-fw1 | TCGTCGGCAGCGTCAGATGTGTATAAGAGACAGNNNNGAKGTRMAGCTTCAGGAGTC |
| IgH-UAd-fw2 | TCGTCGGCAGCGTCAGATGTGTATAAGAGACAGNNNNGAGGTBCAGCTBCAGCAGTC |
| IgH-UAd-fw3 | TCGTCGGCAGCGTCAGATGTGTATAAGAGACAGNNNNCAGGTGCAGCTGAAGSASTC |
| IgH-UAd-fw4 | TCGTCGGCAGCGTCAGATGTGTATAAGAGACAGNNNNGAGGTCCARCTGCAACARTC |
| IgH-UAd-fw5 | TCGTCGGCAGCGTCAGATGTGTATAAGAGACAGNNNNCAGGTYCAGCTBCAGCARTC |
| IgH-UAd-fw6 | TCGTCGGCAGCGTCAGATGTGTATAAGAGACAGNNNNCAGGTYCARCTGCAGCAGTC |
| IgH-UAd-fw7 | TCGTCGGCAGCGTCAGATGTGTATAAGAGACAGNNNNCAGGTCCACGTGAAGCAGTC |
| IgH-UAd-fw8 | TCGTCGGCAGCGTCAGATGTGTATAAGAGACAGNNNNGAGGTGAASSTGGTGGAATC |
| IgH-UAd-fw9 | TCGTCGGCAGCGTCAGATGTGTATAAGAGACAGNNNNGAVGTGAWGYTGGTGGAGTC |
| IgH-UAd-fw10 | TCGTCGGCAGCGTCAGATGTGTATAAGAGACAGNNNNGAGGTGCAGSKGGTGGAGTC |
| IgH-UAd-fw11 | TCGTCGGCAGCGTCAGATGTGTATAAGAGACAGNNNNGAKGTGCAMCTGGTGGAGTC |
| IgH-UAd-fw12 | TCGTCGGCAGCGTCAGATGTGTATAAGAGACAGNNNNGAGGTGAAGCTGATGGARTC |
| IgH-UAd-fw13 | TCGTCGGCAGCGTCAGATGTGTATAAGAGACAGNNNNGAGGTGCARCTTGTTGAGTC |
| IgH-UAd-fw14 | TCGTCGGCAGCGTCAGATGTGTATAAGAGACAGNNNNGARGTRAAGCTTCTCGAGTC |
| IgH-UAd-fw15 | TCGTCGGCAGCGTCAGATGTGTATAAGAGACAGNNNNGAAGTGAARSTTGAGGAGTC |
| IgH-UAd-fw16 | TCGTCGGCAGCGTCAGATGTGTATAAGAGACAGNNNNCAGGTTACTCTRAAAGWGTSTG |
| IgH-UAd-fw17 | TCGTCGGCAGCGTCAGATGTGTATAAGAGACAGNNNNCAGGTCCAACTVCAGCARCC |
| IgH-UAd-fw18 | TCGTCGGCAGCGTCAGATGTGTATAAGAGACAGNNNNGATGTGAACTTGGAAGTGTC |
| IgH-UAd-fw19 | TCGTCGGCAGCGTCAGATGTGTATAAGAGACAGNNNNGAGGTGAAGGTCATCGAGTC |
| **IgH Reverse Primer** | **5′ → 3′ sequence Illumina Adapter sequence read 2 + Diversity region + IgA constant region specific** |
| IgA-const-rev | GTCTCGTGGGCTCGGAGATGTGTATAAGAGACANNNNGAGCTCGTGGGAGTGTCAGTG |

Repertoire overlap was measured by calculating the geometric mean of relative overlap frequencies between CDR3 amino acid sequence usage. The relative overlap similarity was represented on a multi-dimensional scaling (MDS) plot.

**Histology and pathological evaluation of cecum tissue**. For each individual, proximal and distal cecum tissue was embedded in OCT compound (Tissue Tek, DC6994583) and frozen in liquid nitrogen. Three consecutive 6 µm cryosections of each tissue were mounted on glass slides and stained with hematoxylin and eosin following standard protocols. Histopathology was scored in a blinded manner according to the severity of submucosal edema (0–3 score points), the number of polymorphonuclear granulocytes per high-power field in the lamina propria (0–4 score points), reduced numbers of goblet cells (0–3 score points), and epithelial damage (0–3 score points) resulting in a total score of 0–13 points[24]. The mean combined pathological score of proximal and distal cecum is reported.

**Fluorescence microscopy of tissue invaded STm**. Tissue invaded intracellular STm harboring *ssaG::eGFP* reporter plasmid pM973 were visualized and quantified in cecum cryosections prepared from paraformaldehyde-fixed and cryo-embedded cecum tissue as described previously[17]. Sections were stained with DAPI (Sigma, diluted 1:2000) and Phalloidin ATTO 647 (Sigma, diluted 1:500). Up to 12 non-consecutive sections per animal were quantified visually using a Zeiss AXIO Imager.M1 microscope equipped with an EC-Plan-NEOFLUAR 40C/1.3 Oil objective and ×10/23 oculars. One high-power field measures approx. 40,000 µm². Quantitation was carried out in a blinded manner. Images were recorded on a Zeiss LSM710 laser scanning confocal microscope using the Zeiss ZEN 3.1 software. Images were analyzed with the Image J Fiji package.

**mRNA quantification in cecal tissue by qPCR**. Cecum tissue was collected 6 h after infection. Immediately, the tissue was washed in PBS and preserved in RNAlater (Qiagen). The total RNA was extracted from approximately 15 mg tissue, using the RNeasy mini kit (Qiagen). The extraction quality was assessed with Agilent RNA 600 Nano Kit (Qiagen) and reached minimally RIN 9. In total, 5 µg mRNA samples were reversed with RT² easy first strand kit (Qiagen). cDNA libraries were analyzed in a Viia7 Real-Time PCR System and the Viia7 Real-Time PCR System acquisition software (Thermo Scientific) using a RT² profiler PCR array quantifying murine Crohn's disease-related markers (PAMM-169Z, Qiagen) and SYBR green reagents (Qiagen). Five housekeeping genes (Actb, B2m, Gapdh, Gusb, and Hsp90ab1) were averaged and used for calculating ΔCT (=$CT_{sample}-CT_{housekeeping}$). The upper CT limit was fixed to 35 cycles.

**Enzyme-linked immunosorbent assays (ELISA)**. Total lipocalin-2 concentrations of cecum content and fecal pellets were determined by sandwich ELISA using a commercial mouse lipocalin- 2/NGAL ELISA DuoSet (R&D, DY1857), according to the manufacturer's instructions. Immunoglobulin A (IgA) concentrations were quantified from mouse intestinal lavages by sandwich ELISA. ELISA plates were coated with goat anti-mouse IgA (Southern Biotech, 1040-01) and IgA was detected with a horseradish peroxidase (HRP)-conjugated goat anti-mouse IgA (A4789, Sigma). A purified monoclonal IgA isotype antibody (Becton Dickinson, clone M18-254, 553476) served as standard. Absorbance was measured in a 96-microplate reader (VarioskanFlash, version 4.00.53) at 405 nm. Lipocalin-2 and IgA titers were analyzed in Prism 8 for Windows (GraphPad software Inc.). $-EC_{50}$ of each sample/standard was calculated by a four-parameter curve fitting.

**Live bacterial flow cytometry**. Live bacterial flow cytometry quantification of bacterial-specific intestinal IgA titers (expressed as LogEC50 values) were determined as previously described in ref. [25]. Briefly, STm were cultured under SPI1-inducing conditions[67] as described in the Cellular Invasion Assays section. Subsequently, 1 mL of the culture was pelleted at $4816 \times g$ in a Heraeus Fresco 21 centrifuge. The pellet was washed and resuspended to a density of $10^7$ CFU/mL in sterile-filtered 2% BSA/0.005% NaN₃/PBS. Intestinal IgA lavages were collected as described above. Intestinal lavages were serially diluted in sterile-filtered 2% BSA/ 0.005% NaN₃/PBS. Serially diluted Ig-solutions and bacterial suspension were mixed 1:1 and incubated at 4 °C for 1 h. Bacteria were washed twice in sterile-filtered 2% BSA/0.005% NaN3/PBS before re-suspension in monoclonal FITC-anti-mouse IgA (clone 10.3; Becton Dickinson) or PE-anti-mouse IgG1 (clone A85-1; Becton Dickinson) and FITC-anti- mouse IgG2b (clone R12-3; Becton Dickinson). After a further hour of incubation, the bacteria were washed once with PBS/2% BSA/0.005% NaN3/PBS and then resuspended in 2% paraformaldehyde (PFA)/PBS for acquisition on a Becton Dickinson FACSArray SORP or Beckman Coulter Cytoflex S using FSc (forward scatter) and SSc (side scatter) parameters in logarithmic mode. Flow cytometric gating strategy is shown in Fig. S15. Data were analyzed using FlowJo software (Tree Star), and titers were calculated by fitting four-parameter logistic curves[25].

**Statistics**. Statistics were analyzed using Prism 8 for Windows (GraphPad software Inc.). The specific statistical tests used are indicated in the figure legends. Detailed statistical information is provided in the statistical data analysis file available online in the supplementary material.

**Reporting summary**. Further information on research design is available in the Nature Research Reporting Summary linked to this article.

## Data availability

The dataset supporting the conclusions of this article is available as a Source Data file. The raw data underlying Supplementary Fig. 9 (IgA repertoire sequencing) are in the European Nucleotide Archive under ENA accession PRJEB37168 [https://www.ebi.ac.uk/ena/data/view/PRJEB37168]. Preprocessed clonotype amino acid sequences and metadata description are available in the Supplementary Data. All relevant data are available from the authors.

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

## Acknowledgements

We thank Andrew Macpherson for scientific discussions and continued academic support of the gnotobiotic animal infrastructures at University of Bern. We thank staff and management team of the Clean Mouse Facility, DBMR University of Bern, and Elena Verdue, Dana Philpott and the Axenic Gnotobiotic Facility, McMaster University for embryo transfer derivation, maintenance, and sharing of germ free and gnotobiotic mouse lines, Wolf-Dietrich Hardt for providing bacterial strains and plasmids, and Stefanie Buschor, Nicolas Studer, and all other members of the Hapfelmeier laboratory for commenting on the manuscript. Microscopic analyses were supported by the Microscopy Imaging Center (MIC) core facility of the University of Bern. Flow cytometry analyses were supported by the Flow Cytometry Lab core facility of the DMBR, University of Bern. S.H. received funding from the European Research Council (erc.europa.eu) under the European Union's Seventh Framework Programme (FP/2007-2013), ERC Grant Agreement n. 281904 (ERC-2013-StG-281904), and from the Swiss National Science Foundation (www.snf.ch; Grants 310030_138452/1 and 31003A_169791/1), the Helmut-Horten Stiftung (helmut-horten-stiftung.org) and Novartis Foundation (www.stiftungmedbiol.novartis.com; grant 15B120). R.C. was supported by NIH grant NIH R01 AI056289. M.L.B. was supported by SNSF Grant PMPDP3_171261/1 and Novartis Foundation Grant 17C141.

## Author contributions

S.P.P. and O.P.S. performed and analyzed experiments and prepared the figures; M.L.B., A.P., M.D.N. M.M., H.L., J.P.W., M.C., S.H., S.S.U., M.A.T., and F.M.C. performed and analyzed experiments; M.G.A. provided experimental and logistic support; E.S., C.M.S., D.T., and J.P.L. analyzed data; R.C. provided tools and expertise; M.L.B., S.P.P., and O.P.S. analyzed data and co-wrote the manuscript; and S.H. designed and analyzed experiments, acquired funding and wrote the manuscript.

## Competing interests

R.C. III founded Curtiss Healthcare, Inc. in 2015 to develop vaccines to prevent infectious diseases of farm animals. All remaining authors declare no conflict of interest.

**Additional information**

