## [Peer Review File · Nature Communications]

Reviewers' Comments:

Reviewer #1:

Remarks to the Author:

The manuscript by Pfister, Hapfelmeier and colleagues examines the role of inflammation and virulence factor activity in the activation of protective immune responses. This is an interesting and important question. It is quite challenging to dis-entangle the impact of inflammatory pathology itself from the activity of the virulence factors, since the activity is associated with the pathology to begin with. I therefore commend the authors for investigating this question and devising an elegant approach to addressing the distinct contributions of virulence factor activity per se and the pathology triggered by the combined presence of the pathogen and virulence activity. Understanding whether tissue pathology can be separable from virulence during infection has important implications for vaccine design and anti-microbial therapeutic approaches. However, because the entire study is conducted in the setting of germ-free mice, it is not clear whether the conclusions reached here can be broadly applicable. This is a major limitation of the study as it stands. Indeed, the way the title, abstract, and introduction, discussion are written, it appears as though the authors are studying something that happens in a regular (non-germ-free) host. The biggest major overall suggestion I have for this manuscript is that the authors should either do some key studies in a microbiota-sufficient host, to test whether any of the conclusions made here can be broadly applicable (if it is possible to do this), or rewrite the title, abstract, introduction, and conclusions throughout the work to clearly state 'in a germ-free setting'. Currently the conclusions reached in the study cannot be assumed to apply in a setting containing microbiota, and the manuscript does not make that clear.

Specific Points

1. The system devised here, in which bacterial auxotrophy allows mice to be transiently colonized by bacteria that contain virulence factors, is quite elegant. This allows the authors to infect mice with viable wt bacteria or mutants that lack key virulence factors, while limiting bacterial replication and inflammation. Nevertheless, the system appears to require germ free mice since the auxotrophy can be complemented by components of the microbiota which supply a key nutrient required for growth of the mutants. Therefore, while the system is interesting and the conclusions potentially important, it cannot be assumed that the principles described in this manuscript will apply in a host containing an endogenous microbiota. This must be made clear up-front in the abstract and introduction. As written, the abstract, introduction, and discussion make some very broad statements generally about the role of virulence in immunogenicity that I think are not warranted given the limitations of the current study. Nevertheless I think the experiments are well conceived and well-performed, the system itself is quite elegant, and the manuscript overall deserving of publication, if these and the following concerns related to experimental design/robustness can be addressed.

2. Given the very short timeframe of these experiments, it is not appropriate to draw a definitive conclusion about the requirement (or not) of MyD88/Trif in protection (Fig 5). The only timepoint examined here was day 3 post-challenge. Only 5 mice per group are used, and while there is certainly a large drop in bacterial CFU in the immunized MyD88/Trif mice relative to unimmunized mice, it appears as though there are more CFU in the MyD88/Trif mice compared with the wild-type mice (compare red open vs. red closed triangles). This analysis should be done.

3. In general I was unable to find information in figure legends or methods on the number of times an experiment has been repeated. An experiment performed a single time with 5 mice (or even 4 mice as in Casp1/11-/- dataset) is not sufficient to be able to draw robust conclusions. The authors should indicate the number of times and experiment has been done and if the experiments have only been

done once, need to be repeated. The datapoints from multiple experiments should then either be pooled, or the data should be tested for power analysis in order to determine whether the number of datapoints has enough statistical power to be able to draw a conclusion (and how many datapoints are necessary to be able to make a robust interpretation of the data). This applies to both Figs 5 and 6E, in which it looks as though there is in fact elevated CFU burden in the spleen and liver of the Casp1/11^{-/-} mice compared with Casp1/11^{+/-}. Overall I do not find the conclusions involving the knockout mice, particularly the MyD88/Trif, and Casp1/11 mice, to be strongly supported by a robust set of data. The dataset should be expanded and made more robust before any conclusions can really be drawn. The experiments with the salmonella mutants in previous figures have more datapoints, and so this is a little more convincing, but if these have only been done a single time, also need to be repeated.

4. The authors indicate that colonization with wild-type viable but not heat-killed bacteria provides protection against subsequent lethal challenge as measured by bacterial burden and markers of pathology/inflammation on Days 3-4. I realize that the control GF-mice infected with wild-type Salmonella likely have to be euthanized for humane reasons. However, it would be useful to know the extent of protection provided by the auxo-trophic mutants expressing virulence factors at later times if this is possible. (this point is less important than the other two to address, from my perspective)

Reviewer #2:

Remarks to the Author:

Work by Pfister et al is aimed at uncoupling immunogenicity from pathogenicity of invasive enteric bacteria. The researchers wanted to know if exposure to virulent but sterile enteric pathogens were sufficient to generate a protective immune response. They discover that while molecularly intact virulence factors such as the Type III secretion system were important for generating a protective immune response, this immunogenicity could be uncoupled from its ability to grow within the host. Furthermore, they show very nicely how the protective immune response generated as a result of transient exposure to virulent strain was independent of innate immune sensing mechanism but was dependent on activation of adaptive immunity, specifically IgA. Overall this is an elegant and insightful study that describes bacterial virulence and host immune factors are key for generating protective immunity thus setting the scene for developing more efficacious mucosal vaccines for enteric infections. I had the following comments:

1. The role of IgA in providing protective immunity could be explored more. The researcher shows that 4 weeks post-exposure to Aux Salmonella results in robust Stm specific IgA. They further show that Jh^{-/-} mice are unable to generate protective immunity to limit pathogen spread upon challenge with WT Salmonella.
2. How is IgA response upon challenge linked to transient exposure to bacterial virulence factors? Does the ability of Aux strains to infect host cells early on required for generating robust IgA response upon challenge?
3. Are Invasion deficient aux strains unable to trigger Salmonella specific IgA response?
4. Could authors elaborate on the results with PAA-killed bacteria that generates robust IgA response but does not effectively protect against subsequent challenge? The results presented as such establish that IgA is required but other factors (unknown?) are needed to generate protective immunity.
5. In this reviewer's opinion, the flow of the (from the perspective of story-telling) could be improved by putting data eliminating the role of innate immunity (Fig 5 and 6) before data establishing the role of adaptive immunity (fig 2 and 3).
6. In Figure 1A the colors/labels for auxotrophic and non-auxotrophic trains are reversed.

Detailed point-by-point response to the reviewer's comments:

Reviewer #1 (Remarks to the Author):

The manuscript by Pfister, Hapfelmeier and colleagues examines the role of inflammation and virulence factor activity in the activation of protective immune responses. This is an interesting and important question. It is quite challenging to dis-entangle the impact of inflammatory pathology itself from the activity of the virulence factors, since the activity is associated with the pathology to begin with. I therefore commend the authors for investigating this question and devising an elegant approach to addressing the distinct contributions of virulence factor activity per se and the pathology triggered by the combined presence of the pathogen and virulence activity. Understanding whether tissue pathology can be separable from virulence during infection has important implications for vaccine design and anti-microbial therapeutic approaches. However, because the entire study is conducted in the setting of germ-free mice, it is not clear whether the conclusions reached here can be broadly applicable. This is a major limitation of the study as it stands. Indeed, the way the title, abstract, and introduction, discussion are written, it appears as though the authors are studying something that happens in a regular (non-germ-free) host. The biggest major overall suggestion I have for this manuscript is that the authors should either do some key studies in a microbiota-sufficient host, to test whether any of the conclusions made here can be broadly applicable (if it is possible to do this), or rewrite the title, abstract, introduction, and conclusions throughout the work to clearly state 'in a germ-free setting'. Currently the conclusions reached in the study cannot be assumed to apply in a setting containing microbiota, and the manuscript does not make that clear.

Response: We appreciate the reviewer's point that the experiments shown in the first version of the manuscript used germ-free mice. We also agree that the readers should be aware of potential limitations in wider interpretation of results from such models, even though they allow experimental questions to be addressed in a robust way. In this respect we have added caveats as suggested.

See Abstract, line 47-48; Introduction, line 98-100, "*These fundamental studies were carried out in a germ-free setting to avoid the possible confounding factor auxotrophic metabolite cross-feeding by bacteria of the gut microbiota in situ. [...]*"; and Results, line 290-294, "*So far, the fully reversible germ-free mouse model uniquely had allowed the quantitative study of the immunogenicity of different phenotypes of STm^{Aux} in a very clean system. However, in real-life situations STm^{Aux} would interact also with the indigenous gut microbiota, which we hypothesized to provide cross-feeding of the required cell wall metabolites in vivo. This may delay STm^{Aux} intestinal luminal clearance in colonized mice.*".

However, we also took the reviewer's suggestion to experimentally address the possibility that these auxotrophic *Salmonella* strains would be able to survive longer or persist in colonized animals. The new data that has been added to the paper is described below in our response to point 1. Even leveraging this clear data, given that the animals have laboratory levels of colonization diversity, we have been careful over the wider biomedical and veterinary interpretation of potential translatability (Discussion: line 400-403, "However, given that our conclusions so far are based on mouse models that have laboratory levels of microbiota complexity, additional work in more relevant preclinical models will be necessary to assess potential translatability of these findings for veterinary or human medical applications."), although we believe that the results address the general concern raised within our current experimental scope.

Specific Points

1. The system devised here, in which bacterial auxotrophy allows mice to be transiently colonized by bacteria that contain virulence factors, is quite elegant. This allows the authors to infect mice with viable wt bacteria or mutants lacking key virulence factors, while limiting bacterial replication and inflammation. Nevertheless, the system appears to require germ free mice since the auxotrophy can be complemented by components of the microbiota which supply a key nutrient required for growth of the mutants. Therefore, while the system is interesting and the conclusions potentially important, it cannot be assumed that the principles described in this manuscript will apply in a host containing an endogenous microbiota. This must be made clear up-front in the abstract and introduction. As written, the abstract, introduction, and discussion make some very broad statements generally about the role of virulence in immunogenicity that I think are not warranted given the limitations of the current study. Nevertheless I think the experiments are well conceived and well-performed, the system itself is quite elegant, and the manuscript overall deserving of publication, if these and the following concerns related to experimental design/robustness can be addressed.

Response: Thank you for raising this important point. As described in our general response above, in fact we have carried out extensive experiments with auxotrophic *Salmonella* in colonized mice, and have now included these data in the revised manuscript. Notably, crossfeeding of auxotrophs by the microbiota was sufficient not only to delay clearance of STm^{Aux} but indeed allowed normal intestinal luminal colonization. However, it was not sufficient to recover pathogenicity of the auxotroph: Whilst the gut lumen was colonized, almost no tissue invasion and no detectable pathology occurred in either wild-type or MyD88/TRIF-double-deficient colonized mice infected with STm^{Aux}. Moreover, the auxotrophic strain provided robust niche competition to subsequent challenge by wild-type *Salmonella*: a further benefit of such strains in disease prevention. These data are shown in **new Figure 7** and **related supplementary Figures S12-14**.

A further important observation in microbiota-associated mice was that induction of STm-binding specific IgA, as well as disease protection and efficient niche competition were dependent on a functional SPI1 T3SS. Induction of STm-specific IgA by a non-invasive SPI1 T3SS deficient mutant was markedly reduced, despite equivalent colonization levels. Notably, invasive STm^{Aux} showed even lower penetration to mesenteric lymph nodes than a SPI1/SPI2 double deficient avirulent non-auxotrophic strain of STm, highlighting its strong luminal confinement and avirulence.

These additional data support that the uncoupling of virulence factor-driven immunogenicity from pathogenicity by auxotrophic STm is indeed not limited to the germ-free setting. In colonized mice the auxotrophic strain more closely mimics colonization and immune activation of the natural pathogen, yet still effectively uncoupled from the induction of pathology.

The manuscript text has been adjusted (New Results subsection, lines 288-339; Abstract, line 47-50; Introduction, lines 100-104), and the Discussion has been modified (Discussion, lines 394-403) in light of these insightful reviewer comments and our new data.

2. Given the very short timeframe of these experiments, it is not appropriate to draw a definitive conclusion about the requirement (or not) of MyD88/Trif in protection (Fig 5). The only timepoint examined here was day 3 post-challenge. Only 5 mice per group are used, and while there is certainly a large drop in bacterial CFU in the immunized MyD88/Trif mice relative to unimmunized mice, it appears as though there are

Pfister *et al.*, point-by-point response

more CFU in the MyD88/Trif mice compared with the wild-type mice (compare red open vs. red closed triangles). This analysis should be done.

Response: Thank you for highlighting this difficulty. We are not permitted to carry out longer infections in these germ-free immunodeficient mice beyond day 3 on ethical grounds (lethality in this strain at day 4 would be near 100%), and it should be noted that wild-type *Salmonella* can be lethal by this time-point even in wild-type germ-free mice.

We carried out a power analysis (see below: **Table 1. Power analysis**) and calculated that group sizes of n=6-8 would suffice to achieve a power of greater than 90%, and we were able to substantiate our findings by increasing the mouse numbers in the experiments shown in **revised Figure 5** (and related Supplementary **Figure S11**) as recommended. Inclusion of these data confirmed our original observations and supports our previous interpretation. Notably, the extended data set clarified that STm^{Aux} conditioned MYD88/TRIF-deficient mice had no increased cecal mucosal bacterial load compared with the respective wild-type group (**see revised Figure 5 B**), although naïve knockouts had, as expected, a strongly increased mucosal bacterial burden. The microscopic analysis of mucosal STm invasion of all mice depicted in Figure 5 (not only the newly added mice) has been redone in a blinded fashion.

Whilst a survival study beyond day 3 is not permitted in this context under Swiss regulations (these knockout mice tend to succumb to the infection without prior signs of sepsis). However, we would speculate that residual bacterial tissue dissemination and proliferation in immunized MYD88/TRIF deficient mice may impact recovery from challenge infection. However, the time point tested is the latest possible to compare to naïve controls and certainly shows a very strong protective effect of STm^{Aux} induced immunity. We have been careful to point out that the question addressed was the specific role of PRR pathways in induction of adaptive immunity, not their role in innate immune defense or in complementing adaptive immunity in pathogen clearance during later stages of secondary infection. For the latter, they likely play a role, although this role was not yet dominant at day 3 of infection. We adjusted the text results subsection accordingly:

Results, lines 282-286, *“These results show that MYD88/TRIF, Caspase-1/Caspase-11 inflammasome, and NOD1/NOD2 nodosome signaling were individually redundant for the induction of adaptive immunity by live STm^{Aux} in the absence of inflammation. Their role in complementing adaptive immunity in pathogen clearance at later stages of secondary infection is likely functionally important, although not apparent at day 3 of challenge.”*

Abstract, line 43-47, *“Although known to regulate innate immunity to primary *Salmonella* infection, pattern recognition receptor signaling pathways through MYD88 and TRIF, Caspase-1 and -11, NLRC4, and NOD1 and -2 were individually redundant for robust live *Salmonella*-induced mucosal adaptive immunity. This indicated exquisite system redundancy between the inductive mechanisms of anti-pathogen innate and adaptive immunity.”*

3. In general I was unable to find information in figure legends or methods on the number of times an experiment has been repeated. An experiment performed a single time with 5 mice (or even 4 mice as in Casp1/11-/- dataset) is not sufficient to be able to draw robust conclusions. The authors should indicate the number of times and experiment has been done and if the experiments have only been done once, need to be repeated. The datapoints from multiple experiments should then either be pooled, or the data should be tested for power analysis in order to determine whether the number of datapoints has enough statistical power to be able to draw a conclusion (and how many datapoints are necessary to be able to make a robust interpretation of the data). This applies to both Figs 5 and 6E, in which it looks as though there is in fact elevated CFU burden in the spleen and liver of the Casp1/11-/- mice compared with Casp1/11+/- . Overall I do not find the conclusions involving the knockout mice, particularly the MyD88/Trif, and Casp1/11 mice, to be strongly supported by a robust set of data. The dataset should be expanded and made more robust before any conclusions can really be drawn. The experiments with the salmonella mutants in previous figures have more datapoints, and so this is a little more convincing, but if these have only been done a single time, also need to be repeated.

Response: We absolutely agree that reproducibility is very important. We carried out a power analysis as recommended (**see Table 1**). Although sourcing the necessary animal numbers of these rare germ-free mouse lines is highly challenging, we have been able to carry out repeat experiments with additional germ-free Caspase 1/11-/-, and Caspase 1/11 +/- littermates together with NLRC4-/- mice as well as additional NOD1/NOD2 double-deficient mice, and the data were

pooled with the existing data sets, leading to **revised Figure 6**. We find the suggested inclusion of these data considerably strengthened the quality of this part of the paper and confirmed our previous interpretation. As already mention in our discussion of point 2, we pointed out more clearly that these data show that these PRR signaling pathways are individually redundant specifically for the induction of protective adaptive immunity. They certainly are important for innate immunity in primary infection and in complementing adaptive immunity in pathogen clearance during secondary infection, although this role is not yet apparent at day 3 of infection (see revised Results, lines 282-286, and Abstract, lines 43-47)

The experiment shown in revised Figure 4 (T3SS-competent versus -incompetent strains) was carried out in the form of three independent repeats. The data did not scatter according to experimental repeat. According to our power analysis (see Table 1) the animal numbers are adequate.

We apologize for the lack of information in our data reporting in the first version of the manuscript, which has now been corrected. We included details of descriptive statistics and statistical testing, indicated number of independent experiments and groups sizes in the figure legends, and included a statistical summary file as supplementary material. In addition, a source data file underlying all main and supplementary figures will be in the online supplementary information.

Table 1: Power analysis. Calculated according to the “effect size index *f*” method described in chapter 8 of Cohen, J. (1988). Statistical power analysis for the behavioral sciences (2nd ed.). Lawrence Erlbaum Associates.

Power analysis of original submission:

	Fig 4b	Fig 5b	Fig 6c	Fig 6F
Number of groups	5	4	4	6
Current sample size/group	6	5 (approx.)*	6 (approx.)*	4 (approx.)*
Effect size (cohen's d)	0.804	0.968922	0.985034	0.935545
Power (current sample size)*	0.912	0.909	0.968	0.879 ⁺
Significance percentage	0.05	0.05	0.05	0.05

⁺Required 'Power' = 0.9

*Unequal sample size

Power analysis amended dataset:

	Fig 4b	Fig 5b	Fig 6c	Fig 6F
Number of groups	5	4	4	6
Current sample size/group	6	9 (approx.)*	8 (approx.)*	7 (approx.)*
Effect size (cohen's d)	0.8043	0.929368	0.97762	0.974459
Power (current sample size)*	0.912	0.996	0.995	0.998
Significance percentage	0.05	0.05	0.05	0.05

⁺Required 'Power' = 0.9

*Unequal sample size

4. The authors indicate that colonization with wild-type viable but not heat-killed bacteria provides protection against subsequent lethal challenge as measured by bacterial burden and markers of pathology/inflammation on Days 3-4. I realize that the control GF-mice infected with wild-type Salmonella likely have to be euthanized for humane reasons. However, it would be useful to know the extent of protection provided by the auxo-trophic mutants expressing virulence factors at later times if this is possible. (this point is less important than the other two to address, from my perspective)

Response: To address the question whether exposure to live, virulence factor-expressing auxotrophic STm only delays, or improves survival throughout the course of infection, we followed the reviewer's suggestion and carried out an additional experiment where we followed up (and monitored closely the well-being of) live-STm^{Aux}-conditioned germ-free wild-type mice for 3 weeks. This experiment is now included as **new Supplementary Figure S6**. These mice remained free of macroscopic evidence of severe disease, showed beginning improvement of inflammatory markers from day 19, and at the time of sacrifice on day 21 were recovering. It is correct that for ethical reasons and Swiss regulations we cannot allow unprotected control animals to progress beyond day 4 of a wild-type challenge.

For practical reasons and to reduce animal use in adherence to Swiss animal ethical guidelines this experiment was carried out as a combined experiment including also groups of MYD88/TRIF mice that underwent the same pretreatment in the same germ-free isolator. The n = 5 naïve wild type animals sacrificed at day 3 shown in new Figure S6 for comparison served as negative control for both parts of the combined experiment and therefore also appear in revised main Figure 5 in which the MYD88/TRIF mouse experiment data have been included. This has been made transparent in the legend text of Figure S6.

In the light of these data we adjusted the manuscript text by adding the following amendment:

Results, lines 174-178, *“Notably, live-STm^{Aux}-induced immunity not merely delayed the onset of disease, but protected the germ-free mouse from lethal STm infection. Live-STm^{Aux} conditioned germ-free mice that were followed up for 3 weeks following challenge remained free of macroscopic evidence of severe infection and were recovering at the time of sacrifice (Figure S6).”*

Reviewer #2 (Remarks to the Author):

Work by Pfister et al is aimed at uncoupling immunogenicity from pathogenicity of invasive enteric bacteria. The researchers wanted to know if exposure to virulent but sterile enteric pathogens were sufficient to generate a protective immune response. They discover that while molecularly intact virulence factors such as the Type III secretion system were important for generating a protective immune response, this immunogenicity could be uncoupled from its ability to grow within the host. Furthermore, they show very nicely how the protective immune response generated as a result of transient exposure to virulent strain was independent of innate immune sensing mechanism but was dependent on activation of adaptive immunity, specifically IgA. Overall this is an elegant and insightful study that describes that bacterial virulence and host immune factors are key for generating protective immunity thus setting the scene for developing more efficacious mucosal vaccines for enteric infections.

We are very grateful to our colleague for these positive comments.

I had the following comments:

The role of IgA in providing protective immunity could be explored more. The researcher shows that 4 weeks post-exposure to Aux Salmonella results in robust Stm specific IgA. They further show that Jh-/- mice are unable to generate protective immunity to limit pathogen spread upon challenge with WT Salmonella.

1. Could authors elaborate on the results with PAA-killed bacteria that generates robust IgA response but does not effectively protect against subsequent challenge? The results presented as such establish that IgA is required but other factors (unknown?) are needed to generate protective immunity.

Response: We have done additional analysis of the IgA response:

- a. First, we reanalyzed our raw data. The antibody LogEC50 titers presented in the original submitted manuscript were normalized to the antibody concentration in the assay (that is, based on titration curves of antibody binding against antibody concentration). We found this to be misleading, as for example a sample with very low concentrations and a sample with very high concentrations of antibodies of the same specificity would have the same titer, which reflects more the quality of the antibodies rather than the strength of the immune response. We have therefore changed the titer calculation to LogEC50 values not normalized to antibody concentration (that is, based on titration curves of antibody binding versus dilution factor of the sample). For transparency, revised/new main figures 3,4 and 7 are now accompanied with supplemental figures showing the raw IgA binding titration curves (**new Supplementary Figures S5, S8 and S12**).
- b. This data reanalysis revealed that PAA-killed STm preconditioned mice had reduced STm-specific IgA titers at day 1 of challenge (see **revised Figure 3B**). In the **revised Figure 4E** we newly include also day 1 data, showing that both PAA-killed- and T3SS-deficient-STm^{Aux} preconditioned mice had reduced titers at day 1 of challenge. The manuscript text was adjusted accordingly (see Results, lines 169-170, and lines 206-207) Also all other titers shown in the paper have been recalculated to be consistent.
- c. At later time points of the challenge infection (days 3 and 4), when the different treatment had very different levels of intestinal inflammation, STm-specific IgA titer differences were no longer apparent. However, it turned out these similarities were mainly driven by O-antigen specific IgA.

We carried out several experiments to dissect this:

- d. We carried out IgA repertoire analysis based on DNA sequencing of CDR3 regions with the following result:
Results, lines 208-213: "*Immunoglobulin repertoire sequencing analysis of small intestine and mLN revealed overlapping IgA repertoires following mucosal conditioning with live T3SS-competent STm^{Aux}- versus T3SS-incompetent STm^{Aux} that clustered separately from those of naïve germ-free control mice (Figure S9). Repertoire overlap was measured by calculating the geometric mean of relative overlap frequencies between CDR3 amino acid sequence usage...*"

Hence the IgA repertoires were overall rather similar. More detailed methodological information on antibody repertoire data analysis and biocomputational analysis used here will be published in a separate paper (Hai, et al. and Macpherson. Nature, submitted) that we co-author and in which an experiment using T3SS-deficient STm^{Aux} is included. This related manuscript, which is currently under revision at Nature (**manuscript ID 2019-02-02472A**), has been appended to our re-submission.

- e. Next, we carried out an additional experiment in which live STm^{Aux}-preconditioned germ-free mice were challenged with a wild type strain of the different serotype Enteritidis for 3 days, and studied the O-antigen/O-serotype-dependent and -independent reactivity of the STm^{Aux}-induced intestinal IgA. These analyses revealed that virulence factor deficient, as well as killed STm^{Aux} retained the ability to prime O-antigen-specific IgA leading to robust titers against wild-type STm, but were inefficient at inducing IgA of O-serotype independent *Salmonella* surface reactivity. These new results are described in the revised Results section and **revised Figures 4 and S10** as follows:

Results, lines 214-229, "*O-serotype specific IgA has been shown previously to be a necessary component of any intestinal immune protection induced by killed or live STm^{12,26,27}. O-antigen is a dominant repetitive polysaccharide antigen and in binding assays tends to mask other surface epitopes from antibody recognition, which is the basis of O-serotyping. To specifically study O-serotype independent Salmonella surface binding IgA, germ-free mice were pre-conditioned with STm^{Aux} but challenged with the different Salmonella serotype Enteritidis (SEn) (Figure S10 A). The resulting intestinal IgA had reduced surface reactivity towards O-antigen-deficient (rough) STm compared to wild-type (smooth) STm, as expected (Figure S10 B, C). However, the non-O-antigen-specific IgA cross-reacted*

strongly between rough STm and rough SEn (Figure S10 B, C). It also cross-reacted with smooth wild-type SEn, suggesting that it contributes to serotype-independent Salmonella surface reactivity (Figure S10 B, C). Although the O-serotype-independent IgA component alone is insufficient for serotype-independent protection in previously studied mouse models of enteric salmonellosis^{12,26,27}, it may complement O-antigen-specific IgA in protective immunity. Supporting this idea, we found that, although both killed and T3SS-deficient STm^{Aux} pre-conditioning at day 3 of challenge resulted in robust IgA titers towards smooth STm (Figure 4 E, panel Day 3), IgA binding to rough STm was significantly reduced (Figure 4 F).”

- f. It is highly tempting to speculate that improved effector T cell activity is additionally contributing to differential efficacy of mucosal immunity. Although we have some preliminary data in this direction (**see Supplementary Figure S16 for review purposes only**), showing that live STm^{Aux} (and especially, T3SS-competent STm^{Aux}) is a stronger inducer of large intestinal ROR γ t-positive CD4⁺ (Th17) and CD4⁻ TCR γ δ ⁺ T cells, we feel the technical difficulty in dissecting this, places it beyond the realms of this paper. Instead, we adjusted the Discussion text in line 357-358, mentioning this possibility.

2. How is IgA response upon challenge linked to transient exposure to bacterial virulence factors? Is the ability of Aux strains to infect host cells early on required for generating robust IgA response upon challenge?

Response: We carried out additional work supporting this hypothesis:

- a. (see our more detailed response to point 1) We found that, like PAA-killed STm, live T3SS-deficient STm^{Aux} induced less of O-serotype independent surface reactive IgA (see **new panel F in revised Figure 4**).
- b. The data presented in Figure 4 show that strains deficient in SPI1 and SPI2 T3SSs induced immunity of reduced protectiveness and intestinal IgA of reduced O-serotype independent bacterial binding. The SPI1 T3SS is responsible for epithelial invasion and early intestinal pathology. It secretes over a dozen of so-called effector proteins into the target host cell. In an additional mouse experiment (now included in **new supplementary Figure S7**) we show that a triple-effector mutant expressing functional T3SSs, but deficient in the three SPI1 effector proteins SopE, SopE2 and SipA required for cell invasion and early intestinal pathogenesis {Hapfelmeier:2004to} displayed a similarly reduced efficacy as fully T3SS-deficient STm^{Aux}. Hence, the expression of a functional type 3 secretion system (which may to be recognized by innate receptors or raise T3SS specific antibodies) alone is insufficient; the effector protein mediated pathogenic cell signaling is necessary for STm^{Aux} to induce robust immunity. These data are now described in the **Results section (lines 200-205)**. They support the idea that the ability of STm^{Aux} to transiently infect host cells early on is required. The mere display and deployment of functional T3SS apparatuses alone is insufficient.
- c. In **new Figure 7, panel E** (experiments in non-germfree mice; see also our detailed response to Reviewer 1, point 1) we directly demonstrated by microscopy the ability of STm^{Aux} to superficially invade the intestinal epithelium, without deeper penetration or induction of mucosal pathology.

3. Are Invasion deficient aux strains unable to trigger Salmonella specific IgA response?

Response: (See also our responses to the related points 1. and 2.)

- a. As described in more detail above, we find that both noninvasive T3SS mutants and killed STm^{Aux} bacteria may remain robust inducers of STm O-antigen specific IgA. However, they lack the potency of live, T3SS-competent (specifically: SPI1 T3SS competent) STm^{Aux} to induce O-antigen-independent surface binding IgA (see new data in **Figure 4F** and related extended data **new supplemental Figure S8**)
- b. We have included additional experiments in non-germfree mice (see also our response to Reviewer 1, point 1) in **new Figure 7** and related **new Figures S12-14**. In the colonized model, microbiota cross-feeding of STm^{Aux} is sufficient to recover efficient gut luminal colonization, but not mucosal or systemic tissue penetration and pathogenicity. This resulted in a model that more accurately mimics the natural infection and no longer depends on repeated administration of the auxotroph by gavage. Here, induction of wild-type STm-specific IgA depended on invasiveness (SPI1 T3SS competence) of STm^{Aux}. This recapitulates previous findings with live *non*-auxotrophic STm in colonized mice, where protective IgA induction was also found to be T3SS-dependent (but associated with mucosal inflammation), arguing that this model more accurately than the germ-free model recapitulates the natural infection, yet without pathogenesis This information is included in the revised Results section, lines 318-320:

“[...] induction of STm-specific IgA was seen after 4 weeks of colonization, which, in contrast to colonization efficiency, was dependent on SPI1 T3SS-competence (Figure 7G and Figure S12).”

and lines 334-336:

“[...] in the colonized mouse model microbiota-syntrophic STm^{Aux} more closely mimics the intestinal luminal colonization, as well as the virulence factor dependent induction of IgA-mediated immunity of the natural pathogen”.

4. In this reviewer's opinion, the flow of the (from the perspective of story-telling) could be improved by putting data eliminating the role of innate immunity (Fig 5 and 6) before data establishing the role of adaptive immunity (fig 2 and 3).

Response: We thank the reviewer for this suggestion. We tried, but we felt that it is necessary to establish the role of adaptive immunity first, to then dissect the role of innate immune receptor signaling in its induction. Moreover, it is more straight forward to first establish the role of virulence factors in adaptive immune induction, before interrogating the role of innate immune pathways in integration of the immunogenic innate response.

5. In Figure 1A the colors/labels for auxotrophic and non-auxotrophic strains are reversed.

Response: We checked, but were unable to find any inversion of colors/labels.

Reviewers' Comments:

Reviewer #1:

Remarks to the Author:

The authors have thoroughly and carefully addressed the points that were raised during the review. I believe this manuscript will provide an important advance for the field.

Reviewer #2:

Remarks to the Author:

The authors have effectively and diligently addressed all of the comments from the last round of review.

Point-by-point response to referee's comments

Reviewer #1 (Remarks to the Author):

The authors have thoroughly and carefully addressed the points that were raised during the review. I believe this manuscript will provide an important advance for the field.

Reviewer #2 (Remarks to the Author):

The authors have effectively and diligently addressed all of the comments from the last round of review.

We thank both reviewers for the work and scholarship that they put into their feedback.